# Friendship stability in adolescence is associated with ventral striatum responses to vicarious rewards

Elisabeth Schreuders [1,2,3]✉, Barbara R. Braams [4], Eveline A. Crone[1,2,5] & Berna Güroğlu [1,2]

An important task for adolescents is to form and maintain friendships. In this three-wave biannual study, we used a longitudinal neuroscience perspective to examine the dynamics of friendship stability. Relative to childhood and adulthood, adolescence is marked by elevated ventral striatum activity when gaining self-serving rewards. Using a sample of participants between the ages of eight and twenty-eight, we tested age-related changes in ventral striatum response to gaining for stable ($n = 48$) versus unstable best friends ($n = 75$) (and self). In participants with stable friendships, we observed a quadratic developmental trajectory of ventral striatum responses to winning versus losing rewards for friends, whereas participants with unstable best friends showed no age-related changes. Ventral striatum activity in response to winning versus losing for friends further varied with friendship closeness for participants with unstable friendships. We suggest that these findings may reflect changing social motivations related to formation and maintenance of friendships across adolescence.

[1] Department of Developmental and Educational Psychology, Institute of Psychology, Leiden University, Leiden, The Netherlands. [2] Leiden Institute for Brain and Cognition (LIBC), Leiden, The Netherlands. [3] Department of Developmental Psychology, School of Social and Behavioral Sciences, Tilburg University, Tilburg, The Netherlands. [4] Faculty of Behavioural and Movement Sciences, Section Clinical Developmental Psychology, Vrije Universiteit Amsterdam, Amsterdam, The Netherlands. [5] Department of Psychology, Education & Child Studies, Erasmus School of Social and Behavioral Sciences, Erasmus University Rotterdam, Rotterdam, The Netherlands. ✉email: e.schreuders@vu.nl

Adolescence is a transitional period in development during which individuals learn to navigate in an increasing complex social world[1,2]. Friendships—unique relationships that are voluntary and based on equality—become increasingly relevant in adolescence. Not surprisingly, the ventral striatum, a primary reward area, responds to vicarious rewards gained for friends[3]. Relative to childhood and adulthood, adolescence is a period of heightened ventral striatum activity to self-serving rewards (i.e., when gained for the self)[4–7], but not for vicarious rewards for friends[8]. We addressed the question whether different types of friendships affect changes in vicarious reward-related ventral striatum activity across adolescence and young adulthood. Therefore, we conducted a longitudinal study and distinguished between children, adolescents, and adults with two types of best friendships: stable and unstable. Here, participants with stable best friendships had the same best friend at each of the three measurement points (spanning four years), whereas participants with unstable best friendships had a different best friend each time. We examined whether children, adolescents, and young adults with stable and unstable best friendships showed differential ventral striatum responsiveness when rewards are gained for their best friend.

Reward-related ventral striatum activity has been studied extensively[9]. Reward-related ventral striatum activity is shown to relate positively to the immediate pleasure experienced[6,10,11]. Across adolescence, heightened activity in the ventral striatum in response to self-serving rewards has been suggested to play an important role in motivating behaviors, such as pursuing personal goals and novelty seeking[6,12–14]. Furthermore, the ventral striatum is involved in processing rewards in a social context, such as decisions to donate to charity[15], especially while others are watching[16], and when giving money to family[17]. Moreover, ventral striatum responses to vicarious rewards for friends, but not self-serving rewards, related to real-life prosocial behaviors, suggesting that reward sensitivity to vicarious rewards may drive prosocial behavior[18]. There is also evidence of increased activity when sharing gains with friends relative to unfamiliar others[3], and when winning rewards for liked others (i.e., friends) relative to disliked others[19,20]. Braams and Crone[8] examined adolescents' ventral striatum responses to winning for their mother and best friend using cross-sectional data from the current dataset. Ventral striatum activity in response to rewards gained for mothers was heightened in mid-adolescence, echoing the developmental trajectory of ventral striatum sensitivity to rewards for the self. However, ventral striatum activity in response to rewards for friends did not change across adolescence. Together, these findings show that developmental trajectories of reward-related ventral striatum activity are dependent on the social setting, and specifically on the social relationship with the beneficiary.

Friendships not only become more relevant for adolescents, they also become more intimate[21,22] and socially supportive[23–25]. Best friendships are a unique form characterized by high closeness[26,27]. As friendships change through time in response to changing personal needs and circumstances, some best friendships dissolve. It has been reported that about fifty percent of adolescent best friendships are stable throughout one academic year[28,29]. Overall, stable best friendships require more commitment and investment than unstable best friendships and are more common in adolescence than in childhood[29,30].

To extend existing knowledge on reward-driven ventral striatum activity (i.e, winning vs. losing;[6,12]), here, we compared developmental trajectories of ventral striatum responses to vicarious rewards for best friends of adolescents/young adults with unstable versus stable best friendships. To unpack the results further, we examined whether friendship stability relates to differential developmental trajectories of self-serving reward-driven ventral striatum responses characterized by winning vs. losing for self, and interrogated ventral striatum responses to winning/losing for best friends vs. winning/losing for self. In this three-wave biannual longitudinal study, participants of eight to twenty-eight years of age could win or lose money in a heads-or-tails guessing game. First, we tested whether age-related trajectories of ventral striatum activity related to winning vs. losing for friends differed based on friendship stability. We expected that ventral striatum activity would be higher when winning for stable best friends than unstable best friends[3,6,19]. As similarity is a common characteristic of friendships across childhood and adolescence[31], and distinguishes stable friendships from unstable ones[32], individuals with stable friendships may perceive rewards for friends as more similar to rewards for the self than those with unstable friendships. To also understand dynamic relations with changes in friendship quality and friendship closeness we tested whether quality and closeness were positively related to ventral striatum activity when winning vs. losing for friends.

In short, we show that participants with stable best friendships show an age-related quadratic change in ventral striatum activity when rewards were gained vs. lost for best friends with a peak in activity in mid-late adolescence. For participants with unstable best friends no age-related changes were observed, but ventral striatum activity was related positively to friendship closeness. Friendship stability also predicted positive friendship quality and closeness. Together, our findings show that ventral striatum activity in response to vicarious rewards across adolescence is dependent on friendship closeness and relationship stability, with an adolescent-specific neural marker for vicarious gains for stable best friends.

## Results

**ROI approach.** We first examined the whole-brain analysis for the vicarious win > lose contrast for the best friend (see Supplementary Information for details; Supplementary Table 1, Supplementary Fig. 1). Given our a priori hypotheses and confirmed by the findings from our whole-brain analysis, we focused on activation in the nucleus accumbens (NAcc) and examined whether friendship stability modulated age-related changes of NAcc activity, pleasure from winning, friendship quality, and friendship closeness. The fit parameters (AIC & BIC) of the models we tested are listed in Table 1. The parameter estimates and significance of the best model are listed in Table 2 (and Supplementary Table 2). Plots of the raw data are presented as Supplementary Information (Supplementary Fig. 2).

**NAcc activity when winning > losing for friend.** To examine how friendship stability explained variance in ventral striatum activity when winning vs. losing for friends, we conducted separate analyses for the left and right NAcc as outcome variables. We first tested whether sex improved the model fit above and beyond linear and quadratic terms of age to explain variance in NAcc activity. Since there was no main effect of sex or an interaction between age and sex on NAcc activity, sex was removed from the model (left NAcc: $ps > 0.31$; right NAcc: $ps > 0.09$). Next, we tested whether friendship stability improved the model fit. Only a main effect of friendship stability did not improve the model fit (left NAcc: $p = 0.42$; right NAcc: $p = 0.97$). A model that was extended with a friendship stability × age interaction best explained the model (left NAcc: $p = 0.009$, random effects: $SD_{intercept} = 0.22$, $SD_{residual} = 2.31$; right NAcc: $p = 0.008$, random effects: $SD_{intercept} = 0.00$, $SD_{residual} = 2.40$, Fig. 1a, b); there was an interaction between linear age and friendship stability, and between quadratic age and friendship stability (left NAcc: $ps = 0.01$; right NAcc: $ps = 0.009$ and $= 0.02$, respectively).

**Table 1 AIC and BIC values for the models to describe the relation with age, sex, and friendship stability.**

| | df | Left NAcc Friend win > lose | | Right NAcc Friend win > lose | | Pleasure ratings Friend win > lose | |
|---|---|---|---|---|---|---|---|
| | | AIC | BIC | AIC | BIC | AIC | BIC |
| Null | 3 | 1586 | 1597 | 1606 | 1618 | **1761** | **1773** |
| + Linear Age (1) | 4 | 1587 | 1602 | 1608 | 1623 | 1763 | 1778 |
| + Quadratic Age (2) | 5 | 1584 | 1604 | 1608 | 1627 | 1764 | 1783 |
| + Main effect Sex | 6 | 1585 | 1608 | 1607 | 1630 | 1763 | 1786 |
| + Interaction Age and Sex | 8 | 1587 | 1618 | 1609 | 1640 | 1764 | 1795 |
| Age (1 and 2) + Main effect Friendship Stability[a] | 6 | 1586 | 1609 | 1610 | 1633 | 1766 | 1789 |
| + Interaction Age (1 and 2) and Friendship Stability | 8 | **1580** | **1611** | **1604** | **1635** | 1767 | 1798 |

| | df | Positive friendship quality | | Negative friendship quality | | Friendship closeness | |
|---|---|---|---|---|---|---|---|
| | | AIC | BIC | AIC | BIC | AIC | BIC |
| Null | 3 | 354 | 366 | 527 | 539 | 730 | 741 |
| + Linear Age (1) | 4 | 354 | 369 | 519 | 535 | 731 | 744 |
| + Quadratic Age (2) | 5 | 355 | 375 | 520 | 540 | 733 | 750 |
| + Main effect Sex | 6 | *316* | *339* | **518** | **541** | *728* | *748* |
| + Interaction Age and Sex | 8 | 315 | 346 | 522 | 553 | 730 | 757 |
| Age (1 and 2), and Sex effects + Main effect Friendship Stability | 7 | 307 | 335 | 516 | 544 | 727 | 751 |
| + Interaction Sex and Friendship Stability | 8 | **305** | **336** | 518 | 549 | 729 | 756 |
| + Interaction Age (1 and 2) and Friendship Stability | 10 | 308 | 347 | 522 | 561 | **724** | **758** |

*NAcc* nucleus accumbens, *df* degrees of freedom.
Preferred models are shown in bold and effects of sex are shown in italics.
[a]Sex did not improve the model fit and was removed from the model.

To further interrogate the significant interaction between age and friendship stability we performed post hoc tests. These tests revealed that in the stable best friendship group, there is a significant quadratic age effect on NAcc activity (left NAcc: random effects: $SD_{intercept} = 0.00$, $SD_{residual} = 2.09$; fixed effects: [Intercept] $b = 1.68$, $SE = 0.23$, $p < 0.001$; [linear age] $p = 0.07$; [quadratic age] $b = -0.04$, $SE = 0.01$, $p < 0.001$; right NAcc: random effects: $SD_{intercept} = 0.00$, $SD_{residual} = 2.07$; fixed effects: [Intercept] $b = 1.42$, $SE = 0.22$, $p < 0.001$; [linear age] $p = 0.13$; [quadratic age] $b = -0.03$, $SE = 0.01$, $p = 0.002$), whereas there was no significant relation between age and NAcc activity in the unstable best friendship group ($p$s of linear and quadratic age terms $> 0.53$ and $> 0.12$ for left and right NAcc, respectively).

**NAcc activity when winning > losing for self.** To examine how friendship stability explained variance in ventral striatum activity when winning vs. losing for self, we used the same model fitting procedure to test which model best fitted left and right Nacc responses for the contrast between winning and losing for self. The best fitting model for both the left and right NAcc was a model with a quadratic predictor for age. Details are described in the Supplementary Information.

**NAcc activity when winning for friend > self.** Next, we examined how friendship stability explained variance in ventral striatum activity when winning for friend vs. winning for self. This model building procedure showed no significant effects of age, sex, and friendship stability (see Supplementary Information).

**NAcc activity when losing for friend > self.** The model building procedure to examine how friendship stability explained variance in ventral striatum activity when losing for friend vs. losing for self shows a negative effect of linear age on NAcc activity (left NAcc: random effects: $SD_{intercept} = 0.00$, $SD_{residual} = 2.17$;

fixed effects: [Intercept] $b = 0.61$, $SE = 0.12$, $p < 0.001$; [linear age] $b = -0.09$, $SE = 0.03$ $p < 0.01$; right NAcc: random effects: $SD_{intercept} = 0.23$, $SD_{residual} = 2.45$; fixed effects: [Intercept] $b = 0.56$, $SE = 0.13$, $p < 0.001$; [linear age] $b = -0.08$, $SE = 0.04$ $p = 0.03$) (Supplementary Fig. 3). There were no effects of sex and friendship stability (see Supplementary Information for details).

**Vicarious reward-related pleasure ratings.** Next, we tested whether participants with stable and unstable best friendships showed different developmental trajectories of pleasure experienced after winning minus losing (to match the neural contrast). We first tested whether a main effect of sex and an interaction between sex and age improved the model fit for pleasure from winning (versus losing for the best friend) above and beyond linear and quadratic terms of age. These tests showed that there were no significant age-related changes in pleasure ratings ($p$s > 0.41) and there were no main effects of sex and interaction effects of sex with age ($p$s > 0.08). Next, sex was removed from the model, and we tested whether a main effect of friendship stability and an interaction between friendship stability and age significantly improved the model. The results showed that friendship stability was not related to developmental trajectories of pleasure rating of winning for a best friend (model fits $p$s > 0.41), and there was also no main effect of friendship stability (model fits $p$s > 0.19, Fig. 1c).

Post-hoc, we also explored the effect of pleasure from winning and losing for friends separately (i.e., pleasure from winning for friend, and pleasure from losing for friend). These results are described in the supplementary information (Supplementary Table 3). In short, these analyses show no effects of friendship stability on age-related changes of pleasure from winning and losing for friend.

**Friendship quality.** We first built a model including an intercept, a linear term of age, and a quadratic term of age. Then we tested

**Table 2 Statistical parameters, regression coefficients (b), significance level (p) and standard errors (SE) for the bs, for the best fitting mixed-models testing the relation between age and each of the measures reported in the table (two-sided test with α of 0.05).**

| Dependent variable | Fixed effects | b | SE | p |
|---|---|---|---|---|
| Left NAcc win > lose | Intercept | 0.95 | 0.2 | <0.001 |
| | Age, 1 | −0.04 | 0.05 | 0.46 |
| | Age, 2 | 0.00 | 0.01 | 0.80 |
| | Friendship stability | 0.63 | 0.32 | 0.05 |
| | Age, 1 × Friendship stability | 0.19 | 0.08 | 0.01 |
| | Age, 2 × Friendship stability | −0.04 | 0.01 | 0.01 |
| Right NAcc win > lose | Intercept | 1.00 | 0.21 | <0.001 |
| | Age, 1 | −0.09 | 0.05 | 0.10 |
| | Age, 2 | 0.00 | 0.01 | 0.65 |
| | Friendship Stability | 0.38 | 0.33 | 0.24 |
| | Age, 1 × Friendship stability | 0.21 | 0.08 | 0.01 |
| | Age, 2 × Friendship stability | −0.04 | 0.02 | 0.02 |
| Pleasure from winning | Intercept | 4.11 | 0.22 | <0.001 |
| | Age, 1 | −0.01 | 0.05 | 0.80 |
| | Age, 2 | −0.01 | 0.01 | 0.41 |
| Positive friendship quality | Intercept | 4.47 | 0.05 | <0.001 |
| | Age, 1 | 0.01 | 0.01 | 0.25 |
| | Age, 2 | 0.00 | 0.00 | 0.64 |
| | Sex | −0.47 | 0.07 | <0.001 |
| | Friendship stability | 0.08 | 0.07 | 0.29 |
| | Sex × Friendship stability | 0.23 | 0.11 | 0.03 |
| Negative friendship quality | Intercept | 1.69 | 0.05 | <0.001 |
| | Age, 1 | 0.03 | 0.01 | 0.002 |
| | Age, 2 | 0.00 | 0.00 | 0.34 |
| | Sex | 0.15 | 0.08 | 0.05 |
| Friendship closeness | Intercept | 5.33 | 0.17 | <0.001 |
| | Age, 1 | −0.11 | 0.04 | 0.01 |
| | Age, 2 | 0.01 | 0.01 | 0.25 |
| | Sex | −0.62 | 0.23 | 0.01 |
| | Friendship Stability | 0.41 | 0.28 | 0.15 |
| | Age, 1 × Friendship stability | 0.14 | 0.06 | 0.02 |
| | Age, 2 × Friendship stability | −0.03 | 0.01 | 0.05 |
| | Sex × Friendship stability | 0.25 | 0.37 | 0.50 |

*NAcc* nucleus accumbens; *Age*, 1 linear term of *Age*, Age 2 quadratic term of 620 Age, *NAcc win > lose* NAcc activity when winning vs. losing for friend; pleasure from winning = pleasure from winning − losing for friend.

whether a main effect of sex and an interaction between sex and age improved the model fit for friendship quality. Positive friendship quality was best explained by a model including a main effect of sex ($p < 0.001$), and a sex × friendship stability interaction ($p = 0.03$; Random effects: $SD_{intercept} = 0.22$, $SD_{residual} = 0.31$; model fit $p = 0.03$; Fig. 1d). There were no effects of the linear and quadratic terms of age ($ps > 0.25$). Post hoc tests showed that there was a main effect of friendship stability for males (Random effects: $SD_{intercept} = 0.26$, $SD_{residual} = 0.32$; Fixed effects: [intercept] $b = 3.99$, $SE = 0.06$, $p = < .001$; [friendship stability] $b = 0.30$, $SE = 0.09$, $p = 0.002$; [linear age] $p = 0.23$; [quadratic age] $p = 0.62$), such that males with stable best friendships reported higher positive friendship quality than males with unstable best friendships. There was no effect of friendship stability on positive friendship quality for females (random effects: $SD_{intercept} = 0.18$, $SD_{residual} = 0.30$; Fixed effects: [intercept] $b = 4.49$, $SE = 0.04$, $p = < .001$; [friendship stability] $p = 0.15$; [linear age] $p = 0.83$; [quadratic age] $p = 0.10$).

Negative friendship quality was best explained by a linear term of age ($p = 0.003$), and a main effect of sex ($p = 0.0462$; Random effects: $SD_{intercept} = 0.34$, $SD_{residual} = 0.41$; model fit $p = 0.0447$). With increasing age, there was an increase in negative friendship quality and males reported higher levels of negative friendship quality (see Fig. 1e). Friendship stability did not improve the model fit of the developmental trajectory of negative friendship quality ($ps > 0.06$).

**Friendship closeness.** Above and beyond a model with linear and quadratic age terms, a main effect of sex explained additional variance in friendship closeness (model fit: $p = 0.008$). Next, we tested whether main effects of friendship stability and interaction effects with friendship stability improved the model fit. The final model included main effects of the linear age term ($p = 0.008$) and quadratic age term ($p = 25$), a main effect of sex (females > males; $p = 0.008$), a linear age x friendship stability interaction ($p = 0.02$), and a quadratic age x friendship stability interaction, which was significant at trend level ($p = 0.051$; Random effects: $SD_{intercept} = 0.59$, $SD_{residual} = 1.04$; model fit $p = 0.01$). Post hoc tests revealed that there were no age-related changes in friendship closeness for participants with a stable best friendship (random effects: $SD_{intercept} = 0.76$, $SD_{residual} = 1.04$; fixed effects: [intercept] $b = 5.67$, $SE = 0.25$ $p = < 0.001$; [linear age] $p = 0.84$; [quadratic age] $p = 0.13$; [sex] $p = 0.27$), and that friendship closeness decreased linearly with age for participants with an unstable best friendship (random effects: $SD_{intercept} = 0.43$, $SD_{residual} = 1.05$; fixed effects: [intercept] $b = 5.37$, $SE = 0.16$, $p < 0.001$; [linear age] $b = −0.11$, $SE = 0.04$, $p = 0.004$; [quadratic age] $p = 0.25$; [sex] $b = −0.62$, $SE = 0.24$, $p = 0.004$; Fig. 1f).

**Linking NAcc with pleasure, friendship quality, and closeness.** Finally, we examined whether pleasure from winning, friendship quality, and friendship closeness related to NAcc activity when

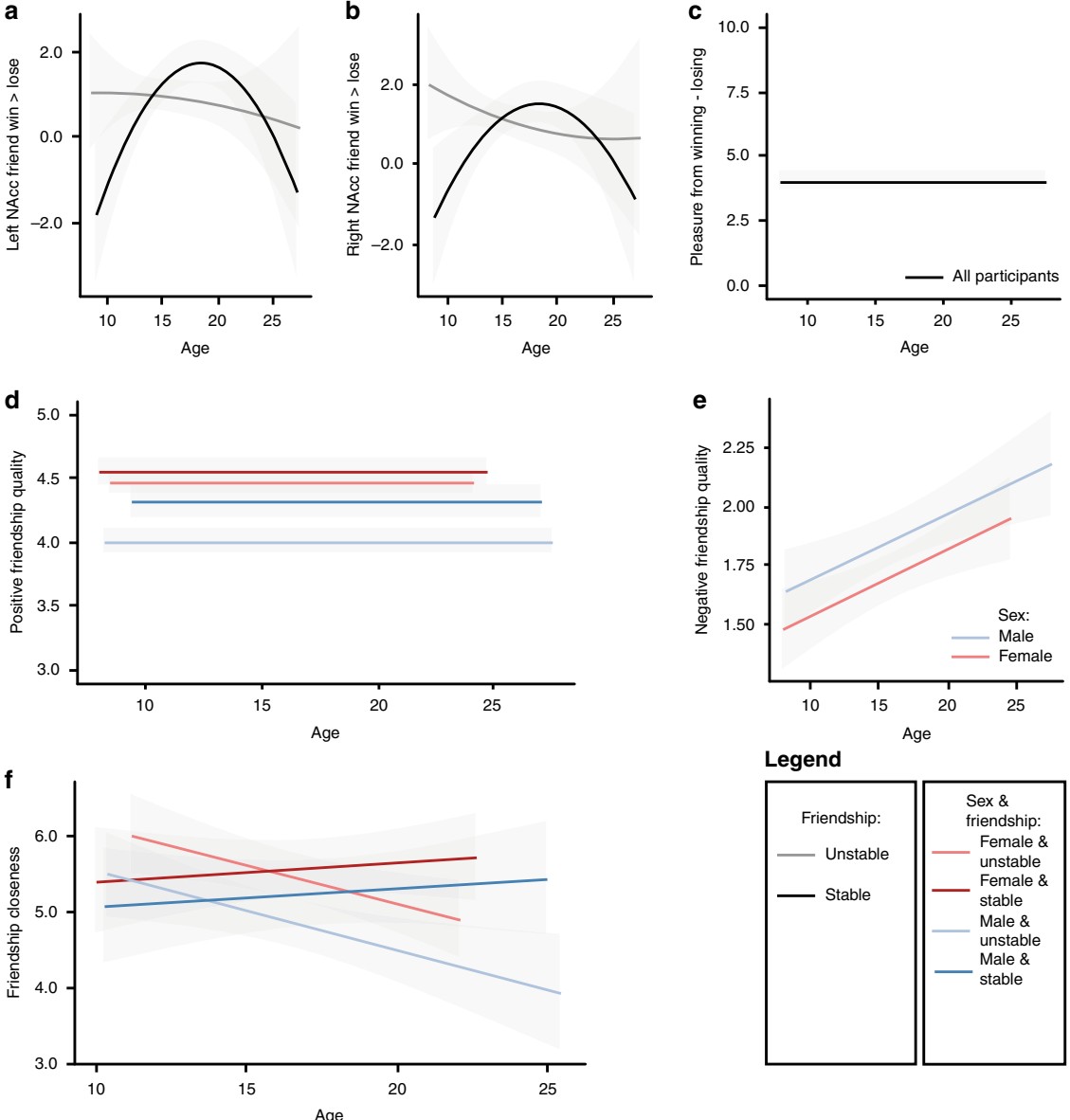

**Fig. 1 Effects of age, sex, and friendship stability.** Data were analyzed with mixed-models with the following outcome variables. **a** Left NAcc activity ($n = 123$, 346 data points). **b** Right NAcc activity ($n = 123$, 346 data points). **c** Pleasure from winning activity ($n = 123$, 363 data points). **d** Positive friendship quality ($n = 123$, 360 data points). **e** Negative friendship quality ($n = 123$, 359 data points). **f** Friendship closeness ($n = 122$, 222 data points). Solid lines represent predicted values of the best fitted model (if not specified in the subpanel, the "legend" explains the color features). The gray ribbon shows the 95% confidence interval.

win > lose for friend. We extracted residuals from the best fitting age models of pleasure from winning versus losing, friendship quality, and friendship closeness to correct for developmental and sex effects in this set of analyses. Furthermore, because our results showed differential age-related trajectories of NAcc activity when winning for a best friend for the stable and unstable friendship groups, we examined the role of pleasure from winning, friendship quality, and closeness on the development of NAcc activity separately for both groups of participants. For participants with stable best friendships, we examined whether a main effect of pleasure from winning versus losing, friendship quality, or closeness explained additional variance above and beyond the linear and quadratic age term. For participants with unstable best friendships, we added a main effect to a model without any age terms, since our results showed no significant age effects on NAcc activity in this group. We conducted separate analyses for each

main effect we tested on NAcc activity (i.e., pleasure from winning, positive and negative friendship quality, and friendship closeness). Table 3 provides an overview of the AIC and BIC parameters for the models we tested.

First, we examined whether ratings of pleasure after winning versus losing for a best friend related to NAcc activity when win > lose for friend. Pleasure from winning versus losing for a best friend was not related to left NAcc activity for neither group of participants with stable (left NAcc: $p = 0.64$; right NAcc: $p = 0.87$) and unstable friendships (left NAcc: $p = 0.86$; right NAcc: $p = 0.44$).

Second, we examined whether friendship quality related to NAcc activity when win > lose for friend. We conducted separate analyses with positive and negative friendship quality as a predictor. Neither positive nor negative friendship quality was related to NAcc activity for participants with stable (left NAcc:

**Table 3 AIC and BIC values for the models to describe relations between NAcc and the pleasure ratings, and friendship quality and closeness.**

| Friendship type & predictor | Left NAcc | | | | Right NAcc | | | |
|---|---|---|---|---|---|---|---|---|
| | Best age model | | + Predictor | | Best age model | | + Predictor | |
| | AIC | BIC | AIC | BIC | AIC | BIC | AIC | BIC |
| Stable best friendship | | | | | | | | |
| Pleasure from winning | **581** | **596** | 583 | 600 | **578** | **592** | 580 | 597 |
| Positive friendship quality | **582** | **597** | 584 | 601 | **580** | **595** | 581 | 599 |
| Negative friendship quality | **582** | **597** | 583 | 600 | **580** | **595** | 582 | 599 |
| Friendship closeness | **366** | **379** | 368 | 383 | **367** | **379** | 368 | 383 |
| Unstable best friendship | | | | | | | | |
| Pleasure from winning | **971** | **981** | 973 | 986 | **994** | **1005** | 996 | 1009 |
| Positive friendship quality | **965** | **975** | 967 | 980 | **990** | **1000** | 992 | 1005 |
| Negative friendship quality | **953** | **963** | 951 | 965 | **981** | **991** | 979 | 993 |
| Friendship closeness | 587 | 596 | **584** | **596** | 598 | 607 | **593** | **605** |

Preferred models are shown in boldm, NAcc nucleus accumbens, NAcc win > lose NAcc activity when winning vs. losing for friend, Pleasure from winning pleasure from winning – losing for friend.

**Unstable best friendships**

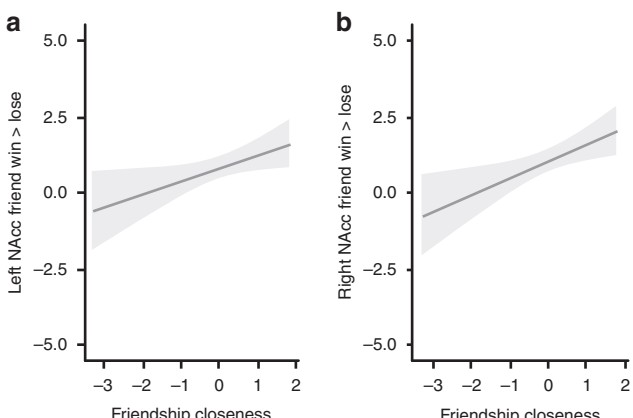

**Fig. 2 Relation between vicarious reward-related NAcc activity and friendship closeness in adolescents with unstable best friendships.** Mixed-models ($n = 74$, 135 data points) were run with the following outcome variables. **a** Left NAcc. **b** Right NAcc. The gray ribbon shows the 95% confidence interval, and the solid line the predicted values.

$p = 0.75$ and 0.20 for positive and negative friendship quality, respectively; right NAcc: $p = 0.37$, and 0.87 for positive and negative friendship quality, respectively) and unstable friendships (left NAcc: $p = 0.84$ and 0.07 for positive and negative friendship quality, respectively; right NAcc: $p = 0.77$ and 0.06 for positive and negative friendship quality, respectively).

Finally, we examined whether friendship closeness related to NAcc activity during win > lose for friend. Self-reported friendship closeness was not related to NAcc activity for participants with stable friendships (left NAcc: $p = 0.69$; right NAcc: $p = 0.55$). There was a significant positive linear relation between friendship closeness and NAcc activity for participants with unstable friendships (left NAcc: $p = 0.03$; right NAcc: $p = 0.009$), such that higher closeness was related to higher NAcc activity (left NAcc: random effects: $SD_{intercept} = 0.55$, $SD_{residual} = 1.97$; fixed effects: [intercept] $b_{intercept} = 0.80$, $SE = 0.18$, $p < 0.001$; [IOS residuals] $b = 0.43$, $SE = 0.19$, $p = 0.03$; right NAcc: random effects: $SD_{intercept} = 0.58$, $SD_{residual} = 2.03$; fixed effects: [intercept] $b_{intercept} = 1.03$, $SE = 0.19$, $p < 0.001$; [IOS residuals] $b = 0.54$, $SE = 0.20$, $p = 0.009$; see Fig. 2 for the fitted relationship and Supplementary Fig. 4 for plot with the raw data).

## Discussion

In this study, we tested whether having stable and unstable best friendships across a trajectory of four years in adolescence was associated with differential developmental trajectories of vicarious reward-related ventral striatum activity. When rewards were gained (vs. lost) for best friends, adolescents with stable best friendships showed a quadratic trajectory of change in ventral striatum activity, whereas adolescents with unstable best friendships showed no age-related changes in their ventral striatum responses to winning for their best friend. Consistent with prior research[6], winning for self related to a quadratic age pattern that was not different for the friendship groups. These effects were not found when directly contrasting winning for friends with winning for self, or losing for friends with losing for self, tentatively suggesting that the effects are driven specifically by differential responses for winning versus losing.

A second important finding was that friendship stability related to positive friendship quality and closeness, confirming that friendship stability is an important aspect of adolescents' friendships. Finally, for participants with unstable best friendships, higher experienced closeness was associated with stronger ventral striatum activity. In the following paragraphs, we set out how friendship stability relates to developmental trajectories of ventral striatum activity and how this scales with friendship quality, friendship closeness, and the pleasure from winning vs. losing.

When comparing winning with losing, our study showed that striatum activity in response to rewards for stable best friends (but not for unstable best friends) followed a peaking quadratic developmental trajectory. Additional analyses showed that participants with stable and unstable best friends showed similar age-related trajectories of ventral striatum activity when rewards were directed to the self (winning vs. losing for self) and when winning/losing for friends and winning/losing for self were compared. Interestingly, the effect of friendship stability on the vicarious reward-related contrast (i.e., winning vs. losing for friends) was not driven by wins or losses (as examined with neural contrasts winning for friend > winning for self, and losing for friend > losing for self), but mainly by the difference in reactivity to wins and losses for best friends. These differential responses should be unpacked further in future studies by including several baseline conditions to examine within-person differences in responses to different feedback schemes. Together, our results confirm our hypothesis that friendship stability is differentially associated with vicarious reward-related activity in the ventral striatum across development.

The increase and decrease in vicarious reward activity in the ventral striatum across adolescence resembles age-related changes of ventral striatum activity when rewards are gained for self[6]. These findings extend prior findings of ventral striatum involvement in processing vicarious rewards for socially close others[19,20]. As such, we show that relationship characteristics may modulate neural sensitivity to outcomes that concern the other person in the relationship. Although these findings do not inform us about the function of stable versus unstable friendships, they show that adolescence may be an important sensitive window for developing stable and close social relationships given the unique adolescent-specific reward response when gaining for stable friends.

We also examined the role of friendship stability on developmental trajectories of friendship closeness, friendship quality, and pleasure from vicarious winning vs. losing. Those participants who reported to have unstable best friendships showed an age-related decrease in closeness with the current best friend extending into adulthood. With a longitudinal, behavioral study on friendship stability, Bowker, Rubin, and Burgess[33] showed that young adolescents (here: around the age of 10 years) with stable best friendships were socially equally well-adjusted as adolescents with unstable best friendships. This highlights the importance of having any best friend in early adolescence. The current study extends these findings by showing that friendship stability becomes more significant across adolescence, and emphasizes that stable friendships may have long-term positive effects on developing close relationships. Furthermore, we showed, for males only, that best friendship quality related to friendship stability such that positive friendship quality was higher for adolescents with stable best friendships than for adolescents with unstable best friendships. There were no differences in pleasure from winning (vs. losing) for the best friend for participants with stable and unstable best friendships. Our findings support previous notions that adolescents and young adults with stable and unstable best friendships differ in their friendship characteristics[34].

Next, we examined how ventral striatum responses to winning as compared to losing for stable and unstable best friends related to friendship quality, friendship closeness, and pleasure from winning. We found that higher ventral striatum activity related to more closeness with the concurrent best friend in participants with unstable best friendships. These results are in line with previous findings suggesting that the closeness of a relationship predicts striatum responses to vicarious rewards[8,35,36]. The findings showing that closeness did not explain individual variation in stable best friend relationships could be due to less variation in closeness experienced in stable best friendship (i.e., mostly high). Our findings suggest that within unstable friendships, vicarious reward sensitivity scales with relative experienced closeness at that time point. Furthermore, ventral striatum activity was not associated with friendship quality and pleasure from winning for both groups of participants with stable and unstable best friendships. Corroborating previous notions that the ventral striatum plays an important role in social re-orientation of adolescents to peers[37], the current study highlights vicarious ventral striatum responsivity as an underlying mechanism for social motivations related to the formation and maintenance of friendships[8]. Future research should further investigate the interplay between striatum responses to rewards for close others and relationship development across adolescence and young adulthood.

Some limitations should be acknowledged. We acknowledge the shortcomings of using losing as a baseline condition when examining vicarious reward responses of the ventral striatum, although winning vs. losing is a commonly used contrast to examine reward sensitivity (for self; e.g., refs. [6,12,38,39]). In an attempt to examine reward sensitivity (i.e., when winning) and punishment sensitivity (i.e., when losing) in isolation, we used the self-reported pleasure ratings to conduct post-hoc analyses. We explored the effect of friendship stability on the developmental trajectory of pleasure from winning and losing for friends and self separately. Our findings show no effects of friendship stability on pleasure from winning or losing for friends. Tentatively, these null findings may also highlight the important role of the relative difference of reactivity to winning and losing.

Furthermore, although a strength of this study was that we used unrestricted nominations of same-sex best friends, we did not incorporate information about friendship duration (before the study started), whether unstable best friends were still part of a close peer network, and whether the best friendships were reciprocated (although the latter is challenging without direct access to the complete peer group). Friendship networks are also subject to significant changes across transition from late adolescence into emerging adulthood as adolescents enter new social networks beyond the classroom structure. Stable friendship that survive such transitional phases (i.e., when continuation of the friendship requires extra effort and investment) might be more likely to be of higher quality than those that dissolve. Future studies should include more information about participants' peer network, including friends and romantic relationships, in order to also obtain a more refined definition of both stable and unstable friendships[34,40]. Furthermore, previous studies showed that similarity and compatibility are important contributors to the continuation of friendships[27,41]. As such, future longitudinal studies could benefit from including the (current) best friend in the study and incorporating information on the friendship dyads. In addition, in future studies, the relation between friendship stability and longitudinal patterns of neural activity and how these relate to mental health of youth could be examined. Finally, future studies using a research paradigm that targets a larger network of brain regions would allow for functional connectivity analyses.

To conclude, our findings show that the developmental trajectory of ventral striatum activity in response to vicarious rewards depends on relationship stability. This suggests that the ventral striatum is involved in tracking social motivations that might influence friendship stability across adolescence. The current study is among the first to provide evidence on how interpersonal peer relationships (here friendships) are related to neural patterns of reward processing and highlight a unique change in vicarious reward experience for stable best friends. Although we did not specifically examine age group differences, the results are consistent with prior studies showing that orientation towards stable friendships does not emerge until mid- to late adolescence[29,37]. Whereas early adolescence might be characterized by social motivation to expand the friend network[21,34,37], mid- to late adolescence might be a crucial period for building stable best friendships. This sets the stage for future studies to examine the links between the development of relationships and neural development across adolescence.

## Methods

**Participants.** The current study is part of a larger, longitudinal study called Braintime, which has been conducted at Leiden University (e.g., see refs. [8,42]) and includes three waves separated by 2 years across a 5-year period. We collected data from 298 healthy, right-handed participants at the first time point (T1), 287 participants at the second time point (T2) and 274 participants at the third time point (T3), resulting in 205 participants that were included in each wave. From this sample, we identified two groups of participants: (a) individuals with a stable best friendship ($n = 48$), and (b) individuals with an unstable best friendship ($n = 75$). To identify these participants, they were asked to name their best friend based on the same question at each time point. Hence, the identification of best friendships

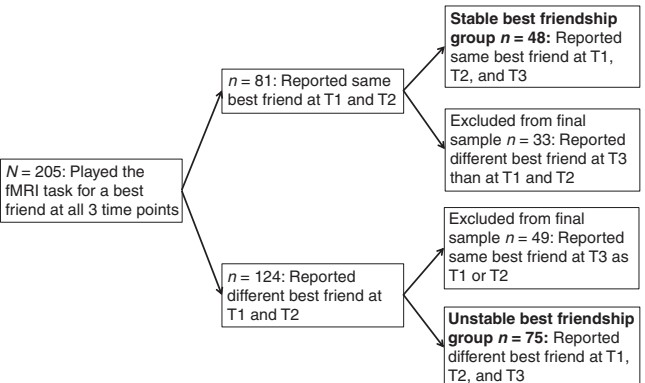

**Fig. 3 Flow chart of participant selection.** We collected data from 298, 287, and 274 participants at the first, second, and third time point, respectively. Of these participants, 205 played the fMRI task for their best friend at all three time points, of which 48 participants with stable best friendships and 75 as participants with unstable best friendships. T1 = time point 1, T2 = time point 2, T3 = time point 3.

was based on unilateral nominations of same-sex best friends at the three time points of data collection; without access to the complete peer group reciprocity of these nominations and duration of friendships beyond the five years were not controlled for. Participants with a stable best friendship named the same best friend at each time point, and participants with an unstable best friendship named a different best friend at each time point (i.e., the best friend at a particular time point was named only once). Those who reported the same best friend at two but not at all three time points were not included because these participants could not easily be categorized as individuals with stable or unstable friendships. Sex was evenly distributed in both the stable and unstable friendship groups ($\chi^2 = 0.13$, $p = 0.72$): there were 28 females with stable friendships (58.3%) and 40 females with unstable friendships (53.3%) (see Fig. 3 for a flow chart of the selection procedure).

Participants in the resulting sample ($N = 123$) were aged 8.01 to 23.44 years at T1 ($M_{age} = 14.11$, $SD = 3.26$), 10.02 to 25.48 years at T2 ($M_{age} = 16.10$, $SD = 3.28$), and 11.95 to 27.54 years at T3 ($M_{age} = 18.11$, $SD = 3.28$). An independent two-sample $t$-test showed that participants with stable friendships were older than participants with unstable friendships (age at T1, stable friendships: $M_{age} = 14.88$ years, $SD = 3.58$; unstable friendships: $M_{age} = 13.62$, $SD = 2.96$; $t(86.56) = -2.04$, $p = 0.04$). Therefore, age was examined as a continuous predictor, and when necessary, controlled for in the analyses. See Fig. 4 for an overview of the age of each participant at each of the time points, where we indicate participants with stable and unstable friendships. The data were inspected for outliers. Extreme outliers were winsorized[43].

**FMRI task**. Functional scans were acquired while participants played a heads-or-tails gambling game in which they had to guess which side of a coin would be chosen by the computer by pressing a button with their right index or middle finger (programmed in E-Prime). Chances of winning on each trial were 50%. The participants started the game with 10 coins. If they guessed correctly, they earned more coins and if they guessed incorrectly, they lost coins (see Fig. 5). Three different types of trials were included in the task to keep the participants engaged: trials on which participants could (a) win 3 or lose 3 coins, (b) win 5 or lose 3 coins, and (c) win 2 or lose 5 coins. Participants were instructed that the coins represented real money, which would be paid out at the end of the experiment. A trial started with a screen showing how many coins could be won or lost (4000 ms) followed by a fixation screen (1000 ms). Next, participants were shown a feedback screen, which revealed whether they won or lost coins (1500 ms). The trial ended with a jittered fixation screen (1000–13200 ms).

At T1 and T2, participants played 30 trials for themselves, 30 trials for their best friend, and 30 trials for another person (disliked peer at T1 and mother at T2). At T3, participants played 23 trials for themselves, and 22 trials for their best friend. This slight change in context is not expected to influence the results for two reasons. First, although the context of the task changed between sessions, the self and friend conditions were presented in a similar way across all sessions, and all participants formed all tasks in the same order. Secondly, NAcc activity across different ages (except the youngest and oldest participants) entails data points from all three time points, and thus all three different experimental designs, because of the accelerated longitudinal nature of the paper. In order to check whether this difference in experimental sessions regarding the 'other trials' (playing for disliked other at T1 and mother at T2) across the three times points affected NAcc responses to winning and losing for friends and the self, we conducted a repeated measures analysis of variance with time point as within subject-factor, sex and friendship stability as between-subject factors, and age at T1 as covariate for NAcc activity in Friend win > lose and Self win > lose. The results show no significant

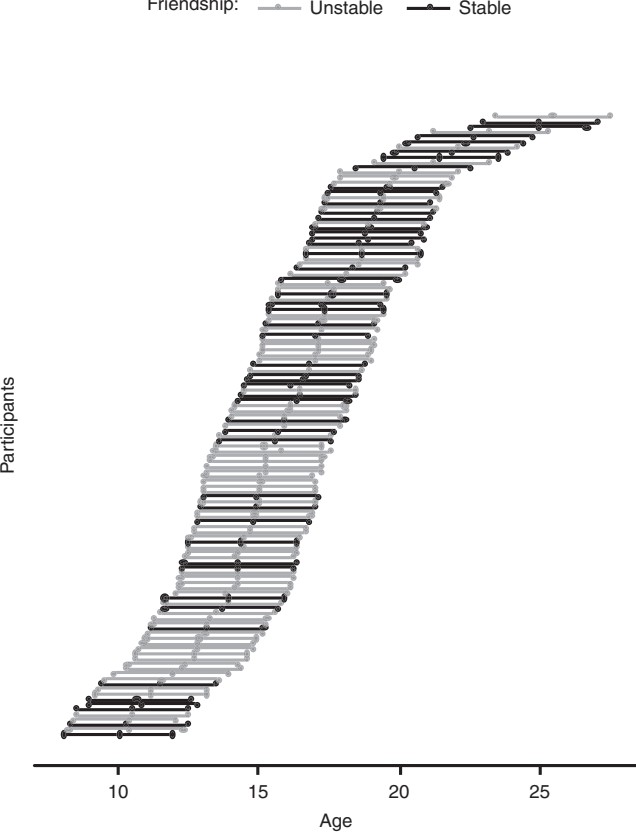

**Fig. 4 Display of age of each participant at three time points of data collection.** Participants with stable friendships are indicated in black ($n = 48$) and with unstable friendships ($n = 75$) in gray. Dots represent a data point and connected dots one participant.

effects of time point on NAcc activity, and are described in detail in the supplementary information. Because our analyses focused on winning and losing for friends and the self, only trials when participants played for their best friends and themselves were included in the analyses.

From the 48 participants with stable best friendships, there were in total 135 valid scan sessions across the three time points that could be used for the analyses (41, 47, and 47 scan sessions obtained at T1, T2, and T3, respectively). Most scan sessions were lost due to excessive motion (motion cut-off > 3 mm movement in any direction) by the participant (six at T2 and one at T3). At T1 one scan session was excluded due to a hole in the functional mask and at T3 one scan session was excluded due to technical problems with the fMRI task. From the 75 participants with unstable best friendships, there were in total 211 valid scan sessions that were used for the analyses (66, 72, and 73 at T1, T2, and T3, respectively). Again, most scans were lost due to excessive motion of the participant during scanning: eight at T1, two at T2, and two at T3. One scan session was lost due to technical difficulties with the fMRI task at T1 and one scan session was excluded due to artifacts at T2.

In the current study, we first conducted the whole-brain analysis on Friend win > lose (see Supplementary Information). Next, we focused on the NAcc activity, a primary reward region in the ventral striatum, during rewards for best friends (i.e., NAcc activity during winning versus losing for friends). We also examined NAcc activity during rewards for self, which is described in detail in the supplementary information. To unpack the win > lose contrasts for friends and self, we also examined the Friend win > Self win and Friend lose > Self lose contrasts.

**MRI data acquisition**. Scans were acquired with a 3 T Philips Achieva MRI scanner. The scanning procedure included (a) a localizer scan, (b) Blood Oxygenation Level Dependent (BOLD) T2* weighted gradient echo planar images (TR = 2.2 s, TE = 30 ms, sequential acquisition, 38 slices of 2.75 mm, field of view (FOV) = 220 mm × 220 mm × 114.7 mm), and (c) an anatomical 3D T1-weighted image (TR = 9.754 ms, TE = 4.59 ms, 8° flip angle, 140 slices, 0.875 mm × 0.875 mm × 1.2 mm, and FOV = 224 mm × 168 mm × 177.3 mm). Two functional runs with 45 trials each were obtained at T1 and T2. At T3, one functional run was obtained in which all 45 trials were presented in the same run. The first two volumes of the functional runs were discarded to allow for equilibration of T1 saturation effects.

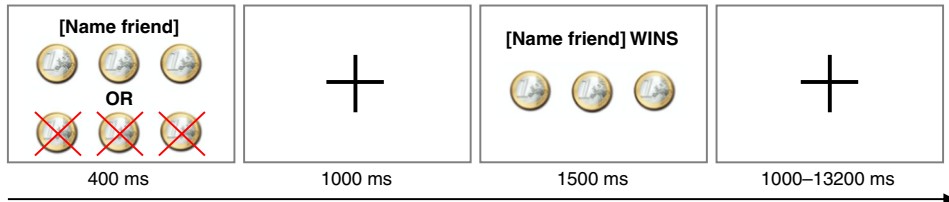

**Fig. 5 Example trial of the fMRI task.** Participants played a gambling task in which they could win or lose money for their best friend. On stimulus onset, a screen was presented showing how much they could win and lose. During the stimulus presentation, participants guessed heads or tails. After a fixation screen, participants received feedback with whether they won or lost for their friend.

| Table 4 Pleasure experienced upon winning and losing for best friend. | | | | |
|---|---|---|---|---|
| **Time point** | **Pleasure from winning** | | **Pleasure from losing** | |
| | **M** | **SD** | **M** | **SD** |
| 1 | 7.67 | 1.87 | 3.53 | 2.19 |
| 2 | 7.34 | 1.61 | 3.60 | 1.98 |
| 3 | 7.39 | 1.35 | 3.19 | 1.37 |
| *M* mean, *SD* standard deviation. | | | | |

**FMRI data analysis**. Neuroimaging data were analyzed using SPM8 software (http://www.fil.ion.ucl.ac.uk/spm/). Preprocessing steps of functional images included realignment, slice-time correction, and smoothing using a Gaussian kernel of 6 mm full-width at half maximum. Functional and structural images were spatially normalized to T1 templates. Templates were based on the Montreal Neurological Institute 305 stereotactic space. Statistical analyses were performed using the general linear model in SPM8. Regressors were modeled as zero-duration events at feedback onset and convolved with a canonical hemodynamic response function. Before conducting the analyses that were targeted at our primary hypotheses, we first examined neural activity in the win > lose contrast when playing for the friend in a 2 [win or lose for friend] × 2 [stable or unstable best friendship] × 3 [T1, T2, or T3] whole-brain ANOVA, FWE corrected, $p < 0.05$, $k \geq 10$ (see Supplementary Information). This analysis was performed to test whether there was activation in other regions than the NAcc when winning vs. losing for friend.

These analyses were followed up by region of interest analyses for longitudinal comparisons. We focused on the NAcc, because this region has been highlighted as a core region in the ventral striatum involved in reward processing[12,13]. We used anatomical masks of the left and right NAcc from the Harvard-Oxford subcortical atlas, thresholded at 40%. These anatomical masks included 28 voxels for the left NAcc and 26 voxels for the right NAcc. The MarsBar toolbox[44] was used to extract the parameter estimates of the left and right NAcc for win > lose for friend left NAcc: intraclass correlation (ICC) = 0.13, 95%−confidence interval (CI) = [−.20, .38]; for T1, T2, and T3, respectively, $M$ = 1.09, 1.28, 0.70, $SD$ = 2.79, 2.79, 1.24; win > lose for friend right NAcc: ICC = −.40, 95%−CI = [−0.43, .26]; for T1, T2, and T3, respectively, $M$ = 1.27, 1.30, 0.70, $SD$ = 2.99, 2.80, 1.18; win > lose for self left NAcc: intraclass correlation (ICC) = 0.22, 95%−CI = [−0.05, 0.44]; for T1, T2, and T3, respectively, $M$ = 2.34, 1.74, 0.97, $SD$ = 2.62, 2.62, 1.10; win > lose for self right NAcc: ICC = 0.04, 95%-CI = [−.30, .30]; for T1, T2, and T3, respectively, $M$ = 2.18, 2.08, 0.94, $SD$ = 2.48, 3.14, 1.24). Two extreme outliers (>3 $SD$s) of the right NAcc for win > lose for friend were winsorized[43].

**Pleasure from winning**. After the MRI session, participants indicated how much pleasure they experienced upon winning and losing for their best friend on an 11-point scale ranging from 0 (not at all) to 10 (really liked winning/losing). For the analyses we used difference scores (pleasure from winning-losing) to keep this measure consistent with the fMRI contrast (NAcc activity during winning > losing for the friend), ICC = 0.59, 95%−CI = [0.45, 0.71]; for T1, T2, and T3, respectively, $M$ = 4.14, 3.73, 4.20 and $SD$ = 3.29, 2.83, 2.33. The means and standard deviations of pleasure from winning and losing for friends are displayed in Table 4. Post-hoc, we also examined the pleasure ratings for pleasure from winning for friend (ICC = 0.61, 95%−CI = [0.47, 0.72]) and pleasure from losing for friend separately (ICC = 0.21, 95%-CI = [−0.07, 0.43])]. The results are reported in the Supplementary Information and briefly discussed in the manuscript.

**Friendship quality**. At T1, T2, and T3, we measured the quality of the relationship with the best friend at each time point using the self-report friendship quality scale (FQS; adapted from refs. [45,46]. Participants indicated on a 5-point scale how true each item was for them from 1 (not true at all) to 5 (very true). Positive friendship quality was measured with 13 questions assessing positive characteristics of the

friendship, like providing support and showing affection (ICC = 0.73 95%−CI = [0.64, 0.81]; for T1, T2, and T3 respectively, Cronbach's α = 0.86, 0.85, 0.75; for T1, T2, and T3, respectively, $M$ = 4.29, 4.35, 4.34 and $SD$ = 0.49, 0.45, 0.36). Higher scores on this scale indicated higher levels of positive friendship quality. Negative friendship quality was measured with seven questions assessing negative characteristics of the friendship, including levels of conflict and power imbalance (ICC = 0.63, 95%−CI = [0.48, 0.74]; for T1, T2, and T3 respectively, Cronbach's α = 0.77, 0.77, 0.64; for T1, T2, and T3, respectively, $M$ = 1.66, 1.69, 1.96 and $SD$ = 0.56, 0.56, 0.41). Higher scores on this scale indicated higher levels of negative friendship quality.

**Friendship closeness**. At T2 and T3, participants indicated how close they felt with their best friend using the inclusion of other in the self (IOS) scale[47]. The IOS scale is a pictorial measure of perceived closeness to others. Participants were instructed to select one picture (of seven in total) that best described the relationship with their best friend. Each of the seven pictures showed two circles: one representing the self and the other one their best friend. The circles in the pictures showed a gradual increase in overlap from picture 1 (circles were not overlapping) to 7 (circles almost entirely overlapping). Thus, a higher proportion of overlap represents a higher level of perceived closeness with the best friend (ICC = 0.45, 95%-CI = [0.19, .63]; for T2 and T3, respectively, $M$ = 5.39, 5.12, and $SD$ = 1.26, 1.25). Correlations between the pleasure from winning, friendship quality, and friendship closeness are reported in the supplementary information (Supplementary Table 4).

**Procedure**. The current study was approved by the Medical Ethics Committee of Leiden University Medical Center (The Netherlands). Participants aged 18 years and older gave written consent for their participation, participants aged 12–17 years gave written assent and their parents provided written consent, and parents from participants under the age of 12 gave written consent for their children's participation. Participants aged 18 years and older received €60 for participation, participants between the ages of 12 and 17 received €30, and participants under the age of 12 received €20. Additionally, all participants could win a small endowment of 3–6 euros for themselves, their best friend or another person when playing the fMRI task[8,42]. Furthermore, participants received 10 (when under the age of 18) or 15 euros (when 18 years of age and older) for filling out additional questionnaires at home.

Before scanning, participants were familiarized with the scanner environment using a mock scanner. They also practiced the fMRI task, in which they could win or lose coins for their best friend. When the experimenter set up the practice run of the task (consisting of 6 trials) for the participants, the participants were asked for the name of their same-sex best friend. This name was used in the practice run as well as during the fMRI-task such that participants saw the name of their best friend when playing for him/her.

**Mixed-model building procedure**. We used a mixed models approach in R[48] for our analyses using the nlme package[49] (and R Studio). We conducted separate tests to examine the main effect of friendship stability and its interaction with age on left and right NAcc activity (friend win > lose, self win > lose, friend win > self win, and friend lose > self lose contrasts), pleasure from winning (friend win > lose), friendship quality, and friendship closeness. A main effect of friendship stability would indicate that participants with stable and unstable best friends score differently on the measure of interest and an interaction between friendship stability and age would indicate that participants with stable and unstable best friends show differential age patterns.

We first built up the model with age and sex predictors. We used the left and right NAcc activity, pleasure from winning, friendship quality, and friendship closeness as dependent variables in the models and added age as a polynomial predictor, and since the data were nested within subjects, we used a random intercept for subjects (also see Braams et al.[12]; Schreuders et al.[6]). We tested for linear and quadratic patterns of age. A linear relation between age and the outcome variable would indicate an age-related increase or decrease. A quadratic relation between age and the outcome variables would indicate a non-linear U or inverted U-pattern. We first built a null model without any predictors, a model with only a linear term of age, and a model with both a linear and quadratic age terms.

Regardless of whether these age terms were significant at this stage of the analyses, we kept them in the model during the model-building procedure to be eventually able to test for interactions between age and friendship stability (and/or sex). Second, we tested whether a main effect of sex and an interaction between age and sex explained additional variance above and beyond the linear and quadratic term of age. Sex was included in the follow-up models if it explained additional variance and excluded if it did not. Sex was dummy coded such that male participants were labeled as 1 and female participants as 0.

Next, we examined whether friendship stability related to left and right NAcc activity, pleasure from winning, friendship quality, and friendship closeness. We tested whether friendship stability explained additional variance in the form of a main effect, and an interaction with age or sex (if sex showed to improve the model fit in previous steps). Friendship stability was dummy coded such that individuals with stable best friendships were labeled as 1 and individuals with unstable best friendships as 0.

Furthermore, in separate models, we tested whether pleasure from winning, friendship quality, and friendship closeness explained additional variance in NAcc activity above and beyond age for participants with stable and unstable best friendships separately. We used the Akaike Information Criterion (AIC[50]; to compare the model fits, and the log likelihood ratio to assess significance. For transparency, we also report the Bayesian Information Criterion (BIC)[51]; we reported the results with a significance threshold of $p < 0.05$.

**Reporting summary**. Further information on research design is available in the Nature Research Reporting Summary linked to this article.

## Data availability
The datasets analyzed during the current study can be found here: https://doi.org/10.17605/OSF.IO/FSYTV. Unthresholded results for main effect of feedback of whole-brain contrast win > lose for friends are available for inspection here: https://neurovault.org/collections/6024/.

## Code availability
Accession codes are available here https://doi.org/10.17605/OSF.IO/FSYTV.

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

## Acknowledgements
The authors thank Anna van Duijvenvoorde, Babette Langeveld, Batsheva Mannheim, Bianca Westhoff, Cédric Koolschijn, Dianne van der Heide, Erik de Water, Jiska Peper, Jochem Spaans, Jorien van Hoorn, Kiki Zanolie, Kyra Lubbers, Laura van der Aar, Mara van der Meulen, Marije Stolte, Neeltje Blankenstein, Rosa Meuwese, Sabine Peters, Sandy Overgaauw, and Suzanne van de Groep for their support during data collection. This work was supported by a European Research Council (ERC) starting grant awarded to Eveline A. Crone (ERC-2010-StG-263234), and a VENI grant from the Netherlands Science Foundation (NWO) awarded to Berna Güroğlu (NWO-VENI 451-10-021).

## Author contributions
E.S. wrote the original draft supervised by B.G. Furthermore, B.R.B., and E.A.C. were involved in reviewing and editing the paper. E.S., E.A.C., and B.G. carried out conceptualization. E.S. and B.R.B. conducted data collection. E.S. carried out the analyses. E.S., B.R.B., E.A.C., and B.G. were involved in data interpretation. E.A.C. and B.G. provided funding.

## Competing interests
The authors declare no competing interests.
