## [Peer Review File · Nature Communications]

Reviewers' comments:

Reviewer #1 (Remarks to the Author):

This is a review of an interesting paper and timely topic examining the development of friendship stability in the brain. The authors take advantage of a large longitudinal data collection at Leiden University and investigate neural responses across 3 time points within a 5 year period. This is the strength of the paper, with a rich data set and unique opportunity to characterize neural signals involved in the development of stable relationships. While there are other strengths with respect to approach and power, there are also some limitations that diminish enthusiasm.

1- The analysis is limited to an ROI approach. It is justified to focus on the ventral striatum ROI given its role in reward and vicarious reward and the authors highlight this in the introduction. However, it seems like a missed opportunity to limit solely to the ventral striatum and not perform a whole brain analysis to probe different brain regions involved in reward or social cognition.

2- The presented analysis are not always straightforward. It is understandable that there are several factors and the authors try to account for all of them using a mixed model approach. At times, statements are made about findings, that perhaps could be corroborated by a simple contrast. For instance, ventral striatum activity was greater when rewards were gained for best friends in mid-to late adolescence but only for individuals with stable friendships. It would seem a contrast could be useful between stable vs non-stable individuals and reward signals during this period.

3- Other potential analyses that could augment the paper would be to better explore the early ventral striatum response (e.g., at T1) and the prediction of stability (vs non-stability) Is there something about individual's striatum signals to vicarious rewards and level of closeness or friendship quality that predicts stability in the future? Further, is the ventral striatum functionally connected with different regions at this time point to strengthen the process?

4- One of the strengths is the longitudinal aspect of data collection. However, this aspect also merits a little more explanation in the exposition. For instance, there are 3 sessions occurring within 5 years. Yet, there are different distribution of age at onset that make it a bit less straightforward. A table with some information about participants (e.g., how many are in the early adolescence group, how many sessions per?) might be helpful. This makes some interpretation difficult as some of the data is from 10-15 year old, while other are from 20-25 year old, and they are grouped together in an analysis where (although age and sex are controlled for) other factors can play a significant role in terms of friendship stability.

5- Overall, the authors have a nice measure of stability (similar best friend for each session vs distinct for each session) and they acknowledge the limitations in the discussion section. It may be helpful to better operationally define stability early in the paper with some of the constraints.

6- The design involves winning (or losing) money for the best friend. In the same task, participants play

trials where they win/lose for a disliked peer (T1), mother (T2) or self (T3). There are several questions here. First, why change the experimental context across the 3 sessions? Second, were trials blocked or mixed in with the best friend trials? Third, the paper suggests that only trials with the friend were accounted for in analysis, but there is a big contextual issue across the 3 tasks. Trials with the disliked peer and mother, for example, should elicit different emotions. Further, reward history, regardless of peer, may impact the next trial and even have a longer carryover effect to a block of trials. Thus, it is difficult to exclude trials from the proposed analysis as the context is different across the 3 sessions. This is a major concern with the current experimental design as the striatum response is sensitive to contextual influences. Finally, why not use the self trials, for example, as a sort of control (it should presumably be the same response regardless of stable or unstable friendships).

7- With respect to the last point, one other question is whether the loss signal, rather than the reward, is driving some of the responses. The current analysis looks at the differential response and is unable to inform the reader with respect to this, but perhaps using the self trials might be helpful.

8- Stability seems to be related to closeness, but in the brain the reward signal associated with closeness was in unstable friendships. This finding seems at odds and would merit more discussion in the paper.

Reviewer #2 (Remarks to the Author):

This study utilizes a 3-wave accelerated longitudinal design to examine how nucleus accumbens activation in response to vicarious rewards for one's best friend change developmentally, and whether this differs among individuals with stable and unstable best friends. While the use of longitudinal data is a major strength of the study, there are concerns that temper my enthusiasm including the focus on age-related patterns rather than mean activation and focusing on NAcc activation to best friend without contrasting to NAcc activation to self.

Line 88: it is not clear from the reviewed literature how it is determined that stable best friendships become more salient in adolescence compared to childhood. Moreover, given that the sample goes up to age 28, a review of how friendships change across the full age range would be helpful.

Hypothesis on line 94 is not well supported. Why would rewards for stable friends peak in adolescence more so than for unstable friends? Please provide greater justification and support for this hypothesis.

It is not clear why the N is much smaller than the full sample. Why is the full sample not included?

It seems problematic that the focus is on age-related patterns of neural activation between the 2 groups, yet age is significantly different between the 2 groups. How do the authors resolve this?

It is not well-justified to focus on just ventral striatum activation on trials for the best friend rather than

contrasting it to the self. As is, results may merely reflect ventral striatum activation (regardless of social actor) that differs between the 2 groups. Contrasting to winning for the self would significantly bolster the findings and underscore that the activation is specific to vicarious rewards.

The focus on age-related patterns (i.e., slope) rather than the mean activation (i.e., intercept) between the groups is not clear. Based on Figure 2A and 2B, it appears that unstable friends have higher overall NAcc activation. This should be unpacked more.

Figure 2A and 2B are difficult to read, as the lines for the stable and unstable friends are nearly the same color. Please differentiate more.

Please be careful of causal language. E.g., it is not evident from the current study that friendship stability affects developmental trajectories of ventral striatum activation (line 408). It could be that the brain affects friendship stability.

The developmental patterns in NAcc activation do not make clear why adolescence is a sensitivity period for developing stable and close relationships. This conclusion is particularly concerning given that NAcc activation appears higher in the unstable friendships, and in the unstable friendships friend closeness is associated with greater NAcc activation. It is difficult to reconcile these findings and conclusions.

Minor: unclear wording throughout making clarity of sentences a little confusing.

line 32 “equal in nature” – not clear what this means

line 33 “adolescents yield reward-related” – word yield is confusing

line 52 “stronger hedonic impact” – what does this mean?

Line 79 “prosperous psychosocial development” – what does prosperous mean here?

Reviewer #3 (Remarks to the Author):

The goal of this study was to examine activation in the ventral striatum in children, adolescent, and adults who vary on friendship stability longitudinally. This is an interesting question and draws on a rich dataset due to its longitudinal design. The methods are sound and the paper is well-written. The inclusion of the raw data in Supplemental information is also commendable. I do have some questions about the interpretation of the study and some methodological questions, as listed below.

1) Friendship instability may have different meaning across development. For example, because adolescence is a time of increased attention to budding friendships, it is developmentally more appropriate to explore friendships with different people at the expense of an exclusive stable best friend. As such, friendship instability in adults may signal problems within the friendship whereas it may signal normative friend exploration and opportunities for psychosocial development in children and

adolescents. As such, categorizing participants into those with and without stable best friendships may not capture friendship quality.

2) How were participants categorized if they named a best friend in two years but not three of the study?

3) What percentage of each age group had stable friendships? The authors note that, unsurprisingly, participants with stable friendships were older than participants with unstable friendships but how imbalanced was this across age groups? This is important information to appropriately interpret the figures showing age-related effects)

4) The authors note that 'there were in total 135 valid scans that could be used for the analyses (41, 47, and 47 scans obtained at T1, T2, and T3, respectively'. Please clarify what "valid scans" refer to. Is each scan considered a volume or a trial or a TR? Does this mean timepoints were censored due to > 3mm motion or that subjects were excluded based on > 3 mm motion?

5) Did the authors examine the data using framewise-displacement?

6) Figure 2a/b is surprising. One might hypothesize the opposite finding (that winning for novel, and perhaps more exciting, friends would elicit greater activation in the NAcc in adolescents given the NAcc's sensitivity to novelty). Can the authors speculate on this?

7) Does NAcc activation exhibit age-related change in this sample when winning for self? (in other words, are the people in each category simply show or less NAcc activity regardless of who they are winning for?)

8) Were any measures collected on mental health--could it be the case that the "unstable" friendship group showed lack of engagement of the adolescent NAcc due to depressive feelings?

9) Were the data examined for outliers? How were outliers in either the questionnaires or brain data handled?

10) Was the best friend question asked the same way each year?

11) Please provide the raw values (not difference scores) for Pleasure for winning as descriptives. Or perhaps you already have? Are these difference score values [M = 4.14, 3.73, 4.20 and SD = 3.26, 2.83, 2.33] and these the raw values [At T1, T2, and T3, pleasure from winning M ranged from 7.34 to 7.67 (SDs = 1.35 to 1.87) and pleasure from losing M ranged from 3.19 to 3.60 (SDs = 1.37 to 2.19)]?

12) How consistent was pleasure for winning > losing ratings across time?

*Changes in the manuscript are blue

Reviewer #1 (Remarks to the Author):

This is a review of an interesting paper and timely topic examining the development of friendship stability in the brain. The authors take advantage of a large longitudinal data collection at Leiden University and investigate neural responses across 3 time points within a 5 year period. This is the strength of the paper, with a rich data set and unique opportunity to characterize neural signals involved in the development of stable relationships. While there are other strengths with respect to approach and power, there are also some limitations that diminish enthusiasm.

1- The analysis is limited to an ROI approach. It is justified to focus on the ventral striatum ROI given its role in reward and vicarious reward and the authors highlight this in the introduction. However, it seems like a missed opportunity to limit solely to the ventral striatum and not perform a whole brain analysis to probe different brain regions involved in reward or social cognition.

We thank the reviewer for making this important suggestion. We report the whole brain analyses in the supplement, and now also briefly describe the results in the manuscript (e.g., p. 8 & 13). As can be seen in Figure S1 and Table S1, the whole brain analysis on vicarious win > lose shows strong and robust activation in the bilateral ventral striatum with peak activity in the bilateral nucleus accumbens. There were no other brain regions that showed significant activations, see also the link to Neurovault where the unthresholded data are available for inspection: <https://neurovault.org/collections/6024/>. This result is consistent with prior research showing that reward activity can be distinguished from other processes in a slow event related design. Given that reward processing was the primary interest in this study, we were convinced that examining the neural developmental patterns for NAcc activity should be our primary focus.

Unfortunately, the current fMRI analyses programs, such as SPM and FSL, do not yet allow for advanced longitudinal modeling at the whole brain level, which is why we extracted the region of interest (ROI) for mixed linear modeling of longitudinal trajectories. This seems a valid approach given the focal activation.

2- The presented analyses are not always straightforward. It is understandable that there are several factors and the authors try to account for all of them using a mixed model approach. At times, statements are made about findings, that perhaps could be corroborated by a simple contrast. For instance, ventral striatum activity was greater when rewards were gained for best friends in mid-to late adolescence but only for individuals with stable friendships. It would seem a contrast could be useful between stable vs non-stable individuals and reward signals during this period.

We thank the reviewer for this suggestion. The advantage of the accelerated longitudinal design is that we make use of the individual data across a 5-year period, with different starting ages to account for practice and cohort effects. This also means that it is not possible to separate age groups (such as young adolescents, or mid adolescents), because participants are part of different age groups at different points in time, and separating these would diminish the longitudinal strengths. However, we completely understand the reviewer's comments

concerning the way the data are presented, and therefore we rewrote sections to make clearly point out that we are testing patterns of linear or quadratic change. We also described more clearly how we contrasted the two groups (stable versus unstable).

P. 11-13: “We used a mixed models approach in R for our analyses (Team, 2014) using the nlme package (Pinheiro, Bates, DebRoy, Sarkar, & Team, 2013). We conducted separate tests to examine the main effect of friendship stability and its interaction with age on left and right NAcc activity, pleasure from winning, friendship quality, and friendship closeness. A main effect of friendship stability would indicate that participants with stable and unstable best friends score differently on the measure of interest and an interaction between friendship stability and age would indicate that participants with stable and unstable best friends show differential age patterns.

We first built up the model with age and sex predictors. We used the left and right NAcc activity, pleasure from winning, friendship quality, and friendship closeness as dependent variables in the models and added age as a polynomial predictor, and since the data were nested within subjects, we used a random intercept for subjects (also see Braams et al., 2015; Schreuders et al., 2018). We tested for linear and quadratic patterns of age. [...]

Next, we examined whether friendship stability related to left and right NAcc activity, pleasure from winning, friendship quality, and friendship closeness. [...]

3- Other potential analyses that could augment the paper would be to better explore the early ventral striatum response (e.g., at T1) and the prediction of stability (vs non-stability) Is there something about individual's striatum signals to vicarious rewards and level of closeness or friendship quality that predicts stability in the future? Further, is the ventral striatum functionally connected with different regions at this time point to strengthen the process?

This is a very interesting suggestion. We tested whether we could predict friendship stability (defined across the three time points) from striatum signal (NAcc activity), pleasure from winning (vs. losing) for friend and friendship quality at T1. We furthermore tested whether friendship stability could be predicted by friendship closeness at T2, because closeness data was not collected at T1. In other words, we used the friendship stability variable that was based on three time points as dependent variable and we used data from the earliest time points for the predictor variables.

In order to control for age and sex effects, we first we first used pleasure, friendship quality, and friendship closeness as dependent variables in separate models and checked whether it was best explained by no age term, a linear age term, or a quadratic age term. After selecting the “best age model” we looked at whether sex explained additional variance. From the best model, if there was an age and/or sex effect, we extracted residuals that were used for the predictor variable for the logistic regressions.

For separate the logistic regressions, we used friendship stability (1 = stable friendship, 0 = unstable friendship) as dependent variable and NAcc activity, pleasure, friendship quality, and friendship closeness collected at the first available time point as independent variables. This resulted in the following model where X is one of the predictor variables corrected for sex and age effects as described above.

Friendship stability <- constant + sex + age + X

All models show a significant effect of age ($bs > .12$, $ps < .04$). None of the predictors of interest (NAcc activity, pleasure, friendship quality, and friendship closeness) significantly predicted friendship stability. However, positive friendship quality and friendship closeness approached significance, indicating that higher scores on positive friendship quality (T1), and higher scores on friendship closeness (T2) might relate to the probability of having a stable best friendship.

Summary of the statistics:

- Left NAcc (T1): $p = .74$
- Right NAcc (T1): $p = .96$
- Pleasure (T1): $p = .49$
- Positive friendship quality (T1):
 - Model $\chi^2(3) = 10.44$, $p = .02$
 - Positive friendship quality: $b = 7.31$, $SE = 3.86$, $p = .06$
- Negative friendship quality (T1): $p = .11$
- Friendship closeness (T2):
 - Model $\chi^2(3) = 8.35$, $p = .04$
 - Friendship closeness: $b = 2.75$, $SE = 1.45$, $p = .06$

The main aim of the current study is to examine the links between friendship stability and striatum responses to vicarious rewards. Considering the non-significant results in predicting friendship stability from striatum responses at earlier time points, we decided to not include these analyses in the current paper, and hope that the reviewer and the editor agree.

The question concerning functional connectivity is highly interesting. The current design was not optimized to test for connectivity patterns because the contrast was developed to separate a specific process (reward processing) and this led to highly localized activity in the bilateral ventral striatum. However, we now suggest in the discussion (p. 21) that in future paradigms it would be interesting to extend this approach to decision-making paradigms that encompass a larger network of brain regions, which would be very well suited for connectivity analyses:

“Finally, future studies using a research paradigm that targets a larger network of brain regions would allow for functional connectivity analyses”

4- One of the strengths is the longitudinal aspect of data collection. However, this aspect also merits a little more explanation in the exposition. For instance, there are 3 sessions occurring within 5 years. Yet, there are different distribution of age at onset that make it a bit less straightforward. A table with some information about participants (e.g., how many are in the early adolescence group, how many sessions per?) might be helpful. This makes some interpretation difficult as some of the data is from 10-15 year old, while other are from 20-25 year old, and they are grouped together in an analysis where (although age and sex are controlled for) other factors can play a significant role in terms of friendship stability.

This comment relates to comment 2 and we agree with the reviewer that this is important to present clearly. We now added a figure (Figure 1) displaying the starting age of each participant and the subsequent testing sessions. We hope this addresses the point of the reviewer well, and we thank the reviewer for making this suggestion. We also made sure that we did not imply anywhere in the text that we separated age periods, as all ages were included

in all analyses. We carefully checked all wording to improve readability and to present the results as clearly as possible.

Figure 2. Display of age of each participant at the three time points of data collection. Participants with stable friendships are indicated in black and with unstable friendships are indicated in grey.

5- Overall, the authors have a nice measure of stability (similar best friend for each session vs distinct for each session) and they acknowledge the limitations in the discussion section. It may be helpful to better operationally define stability early in the paper with some of the constraints.

We thank the reviewer for pointing this out. We added the following information to the methods section (p. 5): “ The identification of best friendships was based on unilateral nominations of same-sex best friends at the three time points of data collection; without access to the complete peer group reciprocity of these nominations and duration of friendships beyond the five years were not controlled for.”

6- The design involves winning (or losing) money for the best friend. In the same task, participants play trials where they win/lose for a disliked peer (T1), mother (T2) or self (T3). There are several questions here. First, why change the experimental context across the 3 sessions? Second, were trials blocked or mixed in with the best friend trials? Third, the paper suggests that only trials with the friend were accounted for in analysis, but there is a big contextual issue across the 3 tasks. Trials with the disliked peer and mother, for example, should elicit different emotions. Further, reward history, regardless of peer, may impact the next trial and even have a longer carryover effect to a block of trials. Thus, it is difficult to exclude trials from the proposed analysis as the context is different across the 3 sessions. This is a major concern with the current experimental design as the striatum response is sensitive to contextual influences. Finally, why not use the self-trials, for example, as a sort of control (it should presumably be the same response regardless of stable or unstable friendships).

These are all very good points. We start with responding to the last question: including the self-trials in the design. This is an excellent suggestion that was also made by the other reviewers (see also comment 5 from reviewer 2 & comment 7 from reviewer 3). We re-analyzed the dataset to also include self-trials at each time point (Self win > lose). This is a strong addition because it allows us to demonstrate the specificity of the effect. Our analyses with the same model fitting procedure testing for NAcc activation for the win>lose contrast show that there is a quadratic effect of age on NAcc activation during winning compared to losing, but that there is no effect of friendship stability. We now include a brief description of these results in the manuscript (p. 14) and details in supplemental information (p.S1-S2).

Supplemental information: “**Neural responses in NAcc when winning > losing for self.** We first tested whether sex explained additional variance beyond a model with a linear factor for age or a model with a quadratic predictor for age. Neither a main effect nor an interaction effect of sex explained additional variance (left NAcc: $p_s > .72$; right NAcc: $p_s > .97$). Sex was therefore removed from the model. We then tested whether friendship stability explained additional variance above the linear and quadratic predictors for age. Neither a main effect nor an interaction effect of friendship stability explained additional variance (left NAcc: $p_s > .25$; right NAcc: $p_s > .47$). The best fitting model for both the left and right NAcc was a model with a quadratic predictor for age (left NAcc: random effects: $SD_{intercept} = .34$, $SD_{residual} = 2.20$; fixed effects: [Intercept] $b = 1.94$, $SE = .15$, $p < .001$; [linear age] $b = -.01$, $SE = .04$, $p = .76$; [quadratic age] $b = -.02$, $SE = .01$, $p = .01$; right NAcc: random effects: $SD_{intercept} = .00$, $SD_{residual} = 2.40$; fixed effects: [Intercept] $b = 2.07$, $SE = .16$, $p < .001$; [linear age] $b = -.04$, $SE = .04$, $p = .30$; [quadratic age] $b = -.03$, $SE = .01$, $p < .001$).”

The reviewer also raises some concern about other potential contextual effects. The reason for changing the conditions between sessions was aimed at avoiding deception (which was involved in the ‘disliked other’ condition used at T1) and we did not want to add this condition on subsequent sessions (i.e., time points T2 and T3). For this reason, disliked other was replaced by mother in the second time point, and there was no third target in the third time point (so T3 included only self and friend trials). Importantly, all participants performed all tasks in the same order. Therefore, contextual effects should be similar for all participants. Nonetheless, we now add this sentence to the discussion to address potential context effects between sessions: “Therefore, the context of the task changed between sessions, but the self and friend conditions were presented in a similar way across all sessions, and all participants formed all tasks in the same order.” (p. 7)

7- With respect to the last point, one other question is whether the loss signal, rather than the reward, is driving some of the responses. The current analysis looks at the differential response and is unable to inform the reader with respect to this, but perhaps using the self-trials might be helpful.

This is a valid point and we do not know for sure whether the effect is driven by larger effects to rewards or smaller effects to loss in mid-adolescence. There was no condition that did not show valence outcomes, because even a no-win condition can be experienced as loss. This makes it almost impossible to have a neutral baseline. Therefore, the results should be interpreted as a relative difference between reward and loss, which is now acknowledged in the discussion (p. 21). It should be noted, however, that there is a large literature implicating the ventral striatum in reward processing, therefore it is likely that the effects in the NAcc are driven by rewards. We also made use of the helpful comments of the reviewer to include the self-trials (see our response to point 6 above).

P. 21: “Furthermore, the neural analyses were based on a difference score between reward and loss trials, therefore, it was not possible to pinpoint whether individual differences are related to differential responses to rewards or losses. Finally, future studies using a research paradigm that targets a larger network of brain regions would allow for functional connectivity analyses.”

8- Stability seems to be related to closeness, but in the brain the reward signal associated with closeness was in unstable friendships. This finding seems at odds and would merit more discussion in the paper.

We thank the reviewer for putting forward this important point. The findings showing that closeness did not explain individual variation in stable best friend relationships is most likely because these friendships are experienced as higher in closeness in general. In contrast, the closeness ratings are on average lower, and therefore most likely more variable, for unstable best friendships. For those participants, within the unstable best friendship group, who report higher closeness, ventral striatum activity for vicarious rewards is also higher. These findings suggest that within unstable friendships, vicarious rewards scale with relative experienced closeness at that time point. This is now described in more detail on page 20: “The findings showing that closeness did not explain individual variation in stable best friend relationships could be due to less variation in closeness experienced in stable best friendship (i.e., mostly high). Our findings suggest that within unstable friendships, vicarious rewards scale with relative experienced closeness at that time point.”

Reviewer #2 (Remarks to the Author):

This study utilizes a 3-wave accelerated longitudinal design to examine how nucleus accumbens activation in response to vicarious rewards for one's best friend change developmentally, and whether this differs among individuals with stable and unstable best friends. While the use of longitudinal data is a major strength of the study, there are concerns that temper my enthusiasm including the focus on age-related patterns rather than mean activation and focusing on NAcc activation to best friend without contrasting to NAcc activation to self.

We thank the reviewer for these important suggestions and both are addressed in the revised version, as outlined below in our reply to the specific points.

1. Line 88: it is not clear from the reviewed literature how it is determined that stable best friendships become more salient in adolescence compared to childhood. Moreover, given that the sample goes up to age 28, a review of how friendships change across the full age range would be helpful.

We thank the reviewer for this excellent point regarding friendship stability across different age groups. Unfortunately, literature on friendship stability across a broad age range, particularly extending into young adulthood is scarce. We now indicate that friendship stability becomes more *common* with increasing age across adolescence. We further refer to the need to investigate friendship stability in the transition from late adolescence to young adulthood in the discussion (p. 21: “Friendship networks are also subject to significant changes across transition from late adolescence into emerging adulthood as adolescents enter new social networks beyond the classroom structure. Stable friendship that survive such transitional phases (i.e., when continuation of the friendship requires extra effort and investment) might be more likely to be of higher quality than those that dissolve.”)

2. Hypothesis on line 94 is not well supported. Why would rewards for stable friends peak in adolescence more so than for unstable friends? Please provide greater justification and support for this hypothesis.

For stable best friends, we expect higher resemblance to self, based on literature reporting higher similarity between best friends and self. This literature is described in more detail on page 4. Therefore, we expected the neural activity pattern to mirror rewards for self. In line with this hypothesis, we now also include the data from self-trials in the supplemental information (p. S1-S2) and a brief description in the manuscript (p. 14). We used the same model building procedure (also including main effects of friendship stability and interactions with friendship stability) to NAcc activity when winning vs. losing for self. Consistent with previous studies, the results show that NAcc activity in response to rewards for self follows a quadratic trajectory (with a peak) across adolescence. Friendship stability did not explain additional variance in this model.

P. 4: “First, we tested whether winning for friend differed based on friendship stability. We expected that ventral striatum activity would be higher when winning for stable best friends than unstable best friends (Fareri et al., 2012; Mobbs et al., 2009), especially in mid adolescence (Schreuders et al., 2018). Similarity is a common characteristic of friendships across childhood and adolescence (Güroğlu, van Lieshout, Haselager, & Scholte, 2007).

Moreover, similarity also distinguishes stable friendships from unstable ones (Hafen, Laursen, Burk, Kerr, & Stattin, 2011). Due to this stronger similarity in stable friendships, individuals with stable friendships may perceive rewards for friends as more similar to rewards for the self than those with unstable friendships. Therefore, we also tested whether responses to rewards for self are similar to responses to rewards for friends.”

Supplemental information: “**Neural responses in NAcc when winning > losing for self.** We first tested whether sex explained additional variance beyond a model with a linear factor for age or a model with a quadratic predictor for age. Neither a main effect nor an interaction effect of sex explained additional variance (left NAcc: $ps > .72$.; right NAcc: $ps > .97$). Sex was therefore removed from the model. We then tested whether friendship stability explained additional variance above the linear and quadratic predictors for age. Neither a main effect nor an interaction effect of friendship stability explained additional variance (left NAcc: $ps > .25$; right NAcc: $ps > .47$). The best fitting model for both the left and right NAcc was a model with a quadratic predictor for age (left NAcc: random effects: $SD_{intercept} = .34$, $SD_{residual} = 2.20$; fixed effects: [Intercept] $b = 1.94$, $SE = .15$, $p < .001$; [linear age] $b = -.01$, $SE = .04$, $p = .76$; [quadratic age] $b = -.02$, $SE = .01$, $p = .01$; right NAcc: random effects: $SD_{intercept} = .00$, $SD_{residual} = 2.40$; fixed effects: [Intercept] $b = 2.07$, $SE = .16$, $p < .001$; [linear age] $b = -.04$, $SE = .04$, $p = .30$; [quadratic age] $b = -.03$, $SE = .01$, $p < .001$).”

3. *It is not clear why the N is much smaller than the full sample. Why is the full sample not included?*

We only included participants who had the same friendship across three timepoints (stable friendship group), and participants who changed friends at each time point (unstable friendship group). Participants who reported the same best friend on two, but not on all three time points were not included because these participants could not easily be categorized as individuals with stable or unstable friendships. We now elaborate more on this categorization on page 5 and we include a flow chart displaying the selection procedure (Figure 1).

p. 5: “To identify these participants, they were asked to name their best friend based on the same question at each time point. Hence, the identification of best friendships was based on unilateral nominations of same-sex best friends at the three time points of data collection; without access to the complete peer group reciprocity of these nominations and duration of friendships beyond the five years were not controlled for. Participants with a stable best friendship named the *same* best friend at *each* time point, and participants with an unstable best friendship named a *different* best friend at *each* time point (i.e., the best friend at a particular time point was named only once). Those who reported the same best friend at two but not at all three time points were not included because these participants could not easily be categorized as individuals with stable or unstable friendships. Sex was evenly distributed in both the stable and unstable friendship groups ($\chi^2 = .13$, $p = .36$): there were 28 females with stable friendships (58.3%) and 40 females with unstable friendships (53.3%) (see Figure 1 for a flow chart of the selection procedure).”

Figure 1. Flow chart of participant selection. We collected data from 298, 287, and 274 participants at the first time point, second time point, and third time point respectively. Of these participants, 205 played the fMRI task for their best friend at all three time points, of which 48 participants were categorized as participants with stable best friendships (same best friend at all three time points) and 75 as participants with unstable best friendships (different best friend at each time point). T1 = time point 1, T2 = time point 2, T3 = time point 3.

4. *It seems problematic that the focus is on age-related patterns of neural activation between the 2 groups, yet age is significantly different between the 2 groups. How do the authors resolve this?*

Even though the mean age is different across the stable and the unstable friendship groups, there are enough participants across the whole age range to test for age-related patterns, see also the new Figure 2 where we display the ages of all participants from the two groups. Given that we control for age in the individual difference analyses, age differences should not affect these effects.

Figure 2. Display of age of each participant at the three time points of data collection. Participants with stable friendships are indicated in black and with unstable friendships are indicated in grey.

5. *It is not well-justified to focus on just ventral striatum activation on trials for the best friend rather than contrasting it to the self. As is, results may merely reflect ventral striatum activation (regardless of social actor) that differs between the 2 groups. Contrasting to winning for the self would significantly bolster the findings and underscore that the activation is specific to vicarious rewards.*

We are grateful to the reviewer for this excellent suggestion. We now also include the analyses for the Self win > lose contrast (p. 14 in the manuscript, and details are described on p. S1-S2 in the supplemental information). The analysis showed a quadratic age effect in the NAcc, corroborating earlier age effects found for winning for the self in this region (Schreuders et al., 2018); there was no main effect of friendship stability, nor an interaction with it.

We also ran analyses comparing NAcc activity related to the Friend win > lose contrast to the Self win > lose contrast. Positive values on this difference measure would indicate that participants showed a larger difference in NAcc response between winning vs. losing for a

friend than for self, whereas negative values would indicate that participants showed a larger difference between winning vs. losing for self. The analysis on the win > lose for friend vs. win > lose for self, yielded an approaching significant main effect of the quadratic age term for the right NAcc ($p = .05$) (not for left NAcc) pointing towards a dip in right NAcc activity (quadratic age term $b = .02$, $SE = .21$, $p = .05$). This could suggest that in mid- to late adolescence, differences in right NAcc response to win > lose for friends and the self are smallest. However, these results should be interpreted with caution as the AIC and BIC values are in opposite directions compared to the model of comparison (see Table below). There was no main effect of friendship stability and no interactions with friendship stability on NAcc activity. We are willing to include these analyses in the supplemental information if the reviewer/editor wants us to.

Table. AIC and BIC values for NAcc activity for (Friend win > lose) > (Self win > lose) to describe the relation with age, sex, and friendship stability. Preferred models are shown in bold and effects of sex are shown in italics. Df = degrees of freedom.

	df	(Friend win > lose) > (Self win > lose)			
		Left Nacc		Right NAcc	
		AIC	BIC	AIC	BIC
Null	3	1746	1757	1781	1793
+ Linear Age (1)	4	1746	1762	1782	1797
+ Quadratic Age (2)	5	1748	1767	1780	1799
+ Main effect Sex	6	1750	1772	1780	1803
+ Interaction Age and Sex	8	1752	1783	1782	1813
Age (1 and 2) + Main effect Friendship Stability ¹	6	1750	1773	1782	1805
+ Interaction Age (1 and 2) and Friendship Stability	8	1750	1781	1782	1813

6. The focus on age-related patterns (i.e., slope) rather than the mean activation (i.e., intercept) between the groups is not clear. Based on Figure 2A and 2B, it appears that unstable friends have higher overall NAcc activation. This should be unpacked more.

Thanks for pointing this out. A main effect of friendship stability is also included in the model building procedure and is only marginally significant for left NAcc ($b = .63$, $p = .05$) and not significant for right NAcc ($p = .24$). These values are presented in Table 3 on page 24. Considering the significant interaction effects between the age terms and friendship stability on NAcc activity, this main effect of friendship stability is less straightforward and needs to be interpreted with caution. Furthermore, we did not test for specific friendship stability main effects within age groups, as the advantage of the longitudinal design is that the same participants are part of different age groups over time (see our detailed reply in comment 2 in reviewer 1 regarding this point).

7. Figure 2A and 2B are difficult to read, as the lines for the stable and unstable friends are nearly the same color. Please differentiate more.

We thank the reviewer for pointing this out. We renewed these figures to improve readability (also following the author guidelines).

Figure 4. Age-related patterns and effects of sex and friendship of A) left NAcc activity, B) right NAcc activity, and C) pleasure from winning, D) positive friendship quality, E) negative friendship quality, and F) friendship closeness. The grey ribbon shows the 95% confidence interval.

8. Please be careful of causal language. E.g., it is not evident from the current study that friendship stability affects developmental trajectories of ventral striatum activation (line 408). It could be that the brain affects friendship stability.

This is an excellent point; we did not want to imply causality and this sentence was rewritten (p. 19 “we set out how friendship stability **relates to** developmental trajectories of ventral striatum activity and how this **scales with** friendship quality, friendship closeness, and the pleasure from winning.”).

9. The developmental patterns in NAcc activation do not make clear why adolescence is a sensitivity period for developing stable and close relationships. This conclusion is a particularly concerning given that NAcc activation appears higher in the unstable friendships, and in the unstable friendships friend closeness is associated with greater NAcc activation. It is difficult to reconcile these findings and conclusions.

We thank the reviewer for pointing out this important issue. We agree that the developmental patterns do not suggest that higher ventral striatum to vicarious rewards is better than lower ventral striatum activity, or that having stable friendships is better than unstable friendships in all phases of adolescence. However, we aimed to point out that adolescence may be an important phase for closeness towards others, and that rewards for stable best friends might be associated with more similarity to rewards for self. Please note that the main effects of friendship stability group are not significant, only the interaction between group x age. This point is now described more clearly on pages 19-20 of the discussion:

“Although these findings do not inform us about the function of stable versus unstable friendships, they show that adolescence may be an important sensitive window for developing stable and close social relationships given the unique adolescent specific reward response when gaining for stable friends.

[...]The findings showing that closeness did not explain individual variation in stable best friend relationships could be due to less variation in closeness experienced in stable best friendship (i.e., mostly high). Our findings suggest that within unstable friendships, vicarious rewards scale with relative experienced closeness at that time point.”

10. Minor: unclear wording throughout making clarity of sentences a little confusing.

We have rewritten all these sentences and thank the reviewer for pointing these out.

line 32 “equal in nature” – not clear what this means: “Based on equality”

line 33 “adolescents yield reward-related” – word yield is confusing: “Not surprisingly, the ventral striatum, a primary reward area, responds to vicarious rewards gained for friends”

line 52 “stronger hedonic impact” – what does this mean?: “Ventral striatum responsiveness to rewards is shown to relate positively to the immediate pleasure experienced”

Line 79 “prosperous psychosocial development” – what does prosperous mean here?: “As such, friendships contribute to adolescents’ psychosocial well-being”

Reviewer #3 (Remarks to the Author):

The goal of this study was to examine activation in the ventral striatum in children, adolescent, and adults who vary on friendship stability longitudinally. This is an interesting question and draws on a rich dataset due to its longitudinal design. The methods are sound and the paper is well-written. The inclusion of the raw data in Supplemental information is also commendable. I do have some questions about the interpretation of the study and some methodological questions, as listed below.

1) Friendship instability may have different meaning across development. For example, because adolescence is a time of increased attention to budding friendships, it is developmentally more appropriate to explore friendships with different people at the expense of an exclusive stable best friend. As such, friendship instability in adults may signal problems within the friendship whereas it may signal normative friend exploration and opportunities for psychosocial development in children and adolescents. As such, categorizing participants into those with and without stable best friendships may not capture friendship quality.

We completely agree with the reviewer that unstable friendships may be associated with exploration and may be beneficial in certain phases of life. To be able to capture developmental changes related to friendship stability across adolescence, we specifically tested whether friendship stability affected the age-related changes of friendship quality and closeness. These analyses show that stable friendships are associated with higher positive friendship quality across adolescence (and do not show differential age patterns). Regarding friendship closeness, adolescents with unstable friendships report a decrease in closeness with age, whereas adolescents with stable friendships report stable closeness with age. These results are included in the manuscript (p. 10). We now discuss the implications of these patterns in more detail on page 19 and 21:

“The current study is among the first to provide evidence on how interpersonal peer relationships (here friendships) are related to neural patterns of reward processing and highlight a unique change in vicarious reward experience for stable best friends. Although we did not specifically examine age group differences, the results are consistent with prior studies showing that orientation towards stable friendships does not emerge until mid- to late adolescence”

“Friendship networks are also subject to significant changes across transition from late adolescence into emerging adulthood as adolescents enter new social networks beyond the classroom structure. Stable friendship that survive such transitional phases (i.e., when continuation of the friendship requires extra effort and investment) might be more likely to be of higher quality than those that dissolve.”

2) How were participants categorized if they named a best friend in two years but not three of the study?

The participants who named a best friend for two sessions and then changed in the third session were not included the study, because that would not allow us to categorize them in one or the other group. We explain this selection now in more detail on page 6 and we include a flow chart for selection of participants (Figure 1).

p. 5: “To identify these participants, they were asked to name their best friend based on the same question at each time point. Hence, the identification of best friendships was based on unilateral nominations of same-sex best friends at the three time points of data collection; without access to the complete peer group reciprocity of these nominations and duration of friendships beyond the five years were not controlled for. Participants with a stable best friendship named the *same* best friend at *each* time point, and participants with an unstable best friendship named a *different* best friend at *each* time point (i.e., the best friend at a particular time point was named only once). Those who reported the same best friend at two but not at all three time points were not included because these participants could not easily be categorized as individuals with stable or unstable friendships. Sex was evenly distributed in both the stable and unstable friendship groups ($\chi^2 = .13, p = .36$): there were 28 females with stable friendships (58.3%) and 40 females with unstable friendships (53.3%) (see Figure 1 for a flow chart of the selection procedure).”

Figure 1. Flow chart of participant selection. We collected data from 298, 287, and 274 participants at the first time point, second time point, and third time point respectively. Of these participants, 205 played the fMRI task for their best friend at all three time points, of which 48 participants were categorized as participants with stable best friendships (same best friend at all three time points) and 75 as participants with unstable best friendships (different best friend at each time point). T1 = time point 1, T2 = time point 2, T3 = time point 3.

3) *What percentage of each age group had stable friendships? The authors note that, unsurprisingly, participants with stable friendships were older than participants with unstable friendships but how imbalanced was this across age groups? This is important information to appropriately interpret the figures showing age-related effects)*

This is an important question, which we addressed in more detail in the revision. We did not test for specific age group effects, as the advantage of the longitudinal design is that the same participants are part of different age groups over time. We now include Figure 2, which presents the ages of all participants at the three time points, separately for the stable and unstable friendship groups. We carefully checked all the language to make sure that we did not imply that separate age groups were analyzed. All ages were represented in both groups; therefore, the mean age effects were not meaningful. In the individual differences analyses, we controlled for age. See also our reply to comment 4 by reviewer 2 on this point.

4) *The authors note that 'there were in total 135 valid scans that could be used for the analyses (41, 47, and 47 scans obtained at T1, T2, and T3, respectively'. Please clarify what "valid scans" refer to. Is each scan considered a volume or a trial or a TR? Does this mean timepoints were censored due to > 3mm motion or that subjects were excluded based on > 3 mm motion?*

We apologize for this misunderstanding. Valid scans referred to valid sessions. We now specified this in the text (p. 7). We excluded all participants who moved more than 3 mm across volumes within a session.

5) *Did the authors examine the data using framewise-displacement?*

We did not use framewise -displacement but instead used the strict criterium of excluding participants completely in case they moved more than 3mm.

6) *Figure 2a/b is surprising. One might hypothesize the opposite finding (that winning for novel, and perhaps more exciting, friends would elicit greater activation in the NAcc in adolescents given the NAcc's sensitivity to novelty). Can the authors speculate on this?*

This is an interesting comment. We do not know whether the 'new' best friend in the unstable friendship group is a completely novel friend, as we did not check whether they recently became friends or knew each other for a longer time. It is difficult to speculate about this, but it is likely that the NAcc response more strongly reflected similarity and closeness than novelty. Therefore, the interesting pattern is for stable friendship participants, who showed a pattern highly similar to winning for self. This is now described on page 4 and 20.

P. 4: "First, we tested whether winning for friend differed based on friendship stability. We expected that ventral striatum activity would be higher when winning for stable best friends than unstable best friends (Fareri et al., 2012; Mobbs et al., 2009), especially in mid adolescence (Schreuders et al., 2018). Similarity is a common characteristic of friendships across childhood and adolescence (Güroğlu, van Lieshout, Haselager, & Scholte, 2007). Moreover, similarity also distinguishes stable friendships from unstable ones (Hafen, Laursen, Burk, Kerr, & Stattin, 2011). Due to this stronger similarity in stable friendships, individuals with stable friendships may perceive rewards for friends as more similar to

rewards for the self than those with unstable friendships. Therefore, we also tested whether responses to rewards for self are similar to responses to rewards for friends.”

P. 19: “Our study showed that striatum activity in response to rewards for stable best friends followed a peaking quadratic developmental trajectory. This confirms our hypothesis that friendship stability is differentially associated with vicarious reward activity in the ventral striatum across development. The increase and decrease in vicarious reward activity in the ventral striatum across adolescence resembles age-related changes of ventral striatum activity when rewards are gained for self (Schreuders et al., 2018). These findings extend prior findings of ventral striatum involvement in processing vicarious rewards for socially close others (Braams et al., 2014a; Mobbs et al., 2009). As such, we show that relationship characteristics may modulate neural responses to outcomes that concern the other person in the relationship. Although these findings do not inform us about the function of stable versus unstable friendships, they show that adolescence may be an important sensitive window for developing stable and close social relationships given the unique adolescent specific reward response when gaining for stable friends.”

7) Does NAcc activation exhibit age-related change in this sample when winning for self? (in other words, are the people in each category simply show or less NAcc activity regardless of who they are winning for?)

We thank the reviewer for pointing this out. We now also include the self-trials in the analyses. The results show a quadratic age effect for winning for self; there was no effect of friendship stability. We now describe these results on page 14 in the manuscript and in the supplemental information on page S1-S2.

Supplemental information: “**Neural responses in NAcc when winning > losing for self.** We first tested whether sex explained additional variance beyond a model with a linear factor for age or a model with a quadratic predictor for age. Neither a main effect nor an interaction effect of sex explained additional variance (left NAcc: $p_s > .72$.; right NAcc: $p_s > .97$). Sex was therefore removed from the model. We then tested whether friendship stability explained additional variance above the linear and quadratic predictors for age. Neither a main effect nor an interaction effect of friendship stability explained additional variance (left NAcc: $p_s > .25$; right NAcc: $p_s > .47$). The best fitting model for both the left and right NAcc was a model with a quadratic predictor for age (left NAcc: random effects: $SD_{intercept} = .34$, $SD_{residual} = 2.20$; fixed effects: [Intercept] $b = 1.94$, $SE = .15$, $p < .001$; [linear age] $b = -.01$, $SE = .04$, $p = .76$; [quadratic age] $b = -.02$, $SE = .01$, $p = .01$; right NAcc: random effects: $SD_{intercept} = .00$, $SD_{residual} = 2.40$; fixed effects: [Intercept] $b = 2.07$, $SE = .16$, $p < .001$; [linear age] $b = -.04$, $SE = .04$, $p = .30$; [quadratic age] $b = -.03$, $SE = .01$, $p < .001$).”

8) Were any measures collected on mental health--could it be the case that the "unstable" friendship group showed lack of engagement of the adolescent NAcc due to depressive feelings?

This would be an excellent follow up question for future research. To already explore whether we could find such relations in our data set, we performed an exploratory analysis that included depression scores (BDI) on time point 3 to test whether stable and unstable

friendships were differentially related to mental health. The results show that BDI scores were not related to group membership, nor to NAcc activity in both groups. We decided not to report these, given that this analysis was post hoc, and the design was not optimized to test these questions. However, we hope that this study inspires other researchers to examine longitudinal patterns and how these relate to mental health in youth (p. 22: “In addition, in future studies, the relation between friendship stability and longitudinal patterns of neural activity and how these relate to mental health of youth could be examined”).

9) *Were the data examined for outliers? How were outliers in either the questionnaires or brain data handled?*

We examined the data for outliers and only extreme outliers were winsorized. This is now indicated on page 6. Only two outliers in right NAcc were winsorized, this is specified on page 9. We carefully checked whether these results were different with and without outliers, but this was not the case.

10) *Was the best friend question asked the same way each year?*

The best friend question was addressed in exactly the same way at each session. This is now specified on page 5: “To identify these participants, they were asked to name their best friend based on the same question at each time point.”

11) *Please provide the raw values (not difference scores) for Pleasure for winning as descriptives. Or perhaps you already have? Are these difference score values [$M = 4.14, 3.73, 4.20$ and $SD = 3.26, 2.83, 2.33$] and these the raw values [At T1, T2, and T3, pleasure from winning M ranged from 7.34 to 7.67 ($SDs = 1.35$ to 1.87) and pleasure from losing M ranged from 3.19 to 3.60 ($SDs = 1.37$ to 2.19)?]*

This is exactly the case, but we now also report them in Table 1 to present these values are clearly as possible.

Table 1. *Pleasure experienced upon winning and losing for best friend. M = mean, SD = standard deviation.*

Time point	Pleasure from winning		Pleasure from losing	
	M	SD	M	SD
1	7.67	1.87	3.56	2.19
2	7.34	1.67	3.60	1.98
3	7.38	1.35	3.19	1.37

12) *How consistent was pleasure for winning > losing ratings across time?*

We tested for consistency by performing ICC analyses, which are described on page 10. The ICC for pleasure from winning > losing for the friend was .59. These show that there is sufficient variation across time.

Reviewers' comments:

Reviewer #1 (Remarks to the Author):

The authors have done a good job with the revision. I am still concerned about just removing the trials at T1 (disliked peer) and T2 (mother) from analysis as they were part of the experimental design and have an effect on responses to win and lose in other conditions. I understand the rationale for changing it across conditions, but it would be useful to demonstrate that those conditions don't significantly impact NAcc responses (e.g., mother vs self trial comparison). Further, the direct comparison between best friend and self should be included and discussed more closely in the main text. I appreciate that the authors find the self trials only change as a function of age, but the direct comparison is also important and should not be minimized. With that being said, the paper has improved and it is interesting and timely with a nice feature of a longitudinal sample.

Reviewer #2 (Remarks to the Author):

The authors have addressed many of my concerns. The paper is interesting, and utilizes a longitudinal design, which is a strength. However, I don't think it is entirely novel or will be of broad interest to the wider field and might be more suitable for a more specialized journal oriented to adolescence or social neuroscience. I also have some concerns remaining about the analyses.

My one major concern remains, and that is the contrast of interest. Like reviewer 1 pointed out, contrasting win>lose for best friend does not allow us to understand whether the neural patterns are driven by winning or losing. Moreover, as I pointed out in my original review, this contrast does not allow us to understand whether the vicarious reward is specific to winning for others or may be reward processing more generally. I would urge the authors to instead focus on win friend > win self, which would appease both these concerns. By separately modeling the win and lose trials, contrasting friend to self, the authors can better unpack whether the effects are driven by winning, losing, and to whom.

Reviewer #3 (Remarks to the Author):

The authors have done a commendable job addressing previous concerns. I remain concerned about the rather ambiguous definitions of friendship stability, closeness and quality but I defer to the other reviewers and editor to make final judgment about whether that ambiguity limits interpretations.

*Changes in the manuscript are in blue

Reviewers' comments:

Reviewer #1 (Remarks to the Author):

The authors have done a good job with the revision. I am still concerned about just removing the trials at T1 (disliked peer) and T2 (mother) from analysis as they were part of the experimental design and have an effect on responses to win and lose in other conditions. I understand the rationale for changing it across conditions, but it would be useful to demonstrate that those conditions don't significantly impact NAcc responses (e.g., mother vs self trial comparison). Further, the direct comparison between best friend and self should be included and discussed more closely in the main text. I appreciate that the authors find the self trials only change as a function of age, but the direct comparison is also important and should not be minimized. With that being said, the paper has improved and it is interesting and timely with a nice feature of a longitudinal sample.

We would like to thank the reviewer for his/her encouraging comments. We now addressed the reviewer's remaining concern about potential context effects of the 'other trials' at T1 (disliked peer) and T2 (mother) on NAcc responses after winning and losing for friends and the self. We conducted a repeated measures ANOVA to test for effects of time point on NAcc responses to winning vs. losing for friends and winning vs. losing for self. One would expect an effect of time point on NAcc activity if including the 'other' trials affected the response to winning and losing for friends and the self. In this repeated measures ANOVA we included time point as the within subject factor (3 levels: T1, T2, and T3), and sex (2 levels: male and female) and friendship stability (2 levels: stable and unstable) as between-subject factors, and added age at T1 as covariate. Separate ANOVAs were conducted for left and right NAcc activity for the Friend win > lose and Self win > lose contrasts (i.e., resulting in 4 ANOVAs, $n = 103$). There was no within-subject effects of time point on NAcc activity during winning vs. losing for friends (left NAcc: main effect of time point $F(2, 97) = .18, p = .84$, interaction effects with time point $ps > .30$; right NAcc: main effect of time point $F(2, 97) = .31, p = .73$, interaction effects with time point $ps > .73$), nor any between-subject effects of sex and friendship stability ($ps > .30$ and $ps > .09$ for left and right NAcc, respectively). Similarly, there was no within-subject effects of time point on NAcc activity during winning vs. losing for self (left NAcc: main effect of time point $F(2, 97) = 1.82, p = .17$, interaction effects with time point $ps > .16$; right NAcc: main effect of time point $F(2, 97) = 2.15, p = .12$, interaction effects with time point $ps > .37$), nor any between-subject effects of sex and friendship stability ($ps > .26$ and $ps > .51$ for left and right NAcc, respectively). Taken together, these analyses suggest that NAcc activity during winning and losing for friends and the self was not affected by changes in the trial procedure across the three time points. We added these analyses to the supplementary information.

Furthermore, we believe that the accelerated longitudinal nature of the paper also partly solves this issue regarding slightly different experimental designs across time points as there are

participants from almost all ages (except for the youngest and oldest participants) at each of the three time points. In other words, NAcc activity across different ages entail data points from all three time points (and thus from all three different experimental designs).

We now emphasized these points in the manuscript (p. 7):

“At T1 and T2, participants played 30 trials for themselves, 30 trials for their best friend, and 30 trials for another person (disliked peer at T1 and mother at T2). At T3, participants played 23 trials for themselves, and 22 trials for their best friend. This slight change in context is not expected to influence the results for two reasons. First, although the context of the task changed between sessions, the self and friend conditions were presented in a similar way across all sessions, and all participants formed all tasks in the same order. Secondly, NAcc activity across different ages (except the youngest and oldest participants) entails data points from all three time points, and thus all three different experimental designs, because of the accelerated longitudinal nature of the paper. In order to check whether this difference in experimental sessions regarding the ‘other trials’ (playing for *disliked other* at T1 and *mother* at T2) across the three times points affected NAcc responses to winning and losing for friends and the self, we conducted a repeated measures analysis of variance with time point as within subject-factor, sex and friendship stability as between-subject factors, and age at T1 as covariate for NAcc activity in Friend win > lose and Self win > lose. The results show no significant effects of time point on NAcc activity, and are described in detail in the supplementary information. Because our analyses focused on winning and losing for friends and the self, only trials when participants played for their best friends and themselves were included in the analyses.”

The reviewer also encourages us to add a friend vs. self comparison in the manuscript. In line with the comments of reviewer 2, we now also report the Friend win > Self win and Friend lose > Self lose contrasts in the manuscript. Please see our response below in our reply to Reviewer2 regarding this point.

Reviewer #2 (Remarks to the Author):

The authors have addressed many of my concerns. The paper is interesting, and utilizes a longitudinal design, which is a strength. However, I don't think it is entirely novel or will be of broad interest to the wider field and might be more suitable for a more specialized journal oriented to adolescence or social neuroscience. I also have some concerns remaining about the analyses.

My one major concern remains, and that is the contrast of interest. Like reviewer 1 pointed out, contrasting win>lose for best friend does not allow us to understand whether the neural patterns are driven by winning or losing. Moreover, as I pointed out in my original review, this contrast does not allow us to understand whether the vicarious reward is specific to winning for others or may be reward processing more generally. I would urge the authors to instead focus on win friend > win self, which would appease both these concerns. By separately modeling the win and

lose trials, contrasting friend to self, the authors can better unpack whether the effects are driven by winning, losing, and to whom.

We thank the reviewer for his/her enthusiasm. With the current study, we aimed to extend the existing knowledge on win vs. lose reward-related NAcc activity for a different beneficiary, and also based our hypotheses on previous work reporting this contrast (Schreuders et al., 2018; Braams et al., 2014a; 2014b; 2015; 2017). However, we understand the reviewer's concern and now also report the Friend win > Self win and Friend lose > Self lose contrasts and discuss the results. Friendship stability was found not be related to NAcc activity in both contrasts. There was only a negative linear age effect for NAcc activity related to the Friend lose > Self lose contrast. We now incorporated this in the manuscript as follows:

Introduction p. 4: "To extend existing knowledge on reward driven NAcc activation (i.e, winning vs. losing; Braams et al., 2015; Schreuders et al., 2018), here, we compared developmental trajectories of purely vicarious reward-driven response of the ventral striatum for best friends of adolescents/young adults with unstable versus stable best friendships. To unpack the results of this analysis, we also examined whether friendship stability relates to differential developmental trajectories of self-directed, reward-driven response of the ventral striatum characterized by winning vs. losing for self, and interrogated ventral striatum responses to winning/losing for best friends vs. winning/losing for self."

Method section p. 8: "In the current study, we first examined the whole brain analysis on Friend win > lose (see supplementary information). Next, we focused on the nucleus accumbens (NAcc) activity, a primary reward region in the ventral striatum, during rewards for best friends (i.e., NAcc activity during winning versus losing for friends). We also examined NAcc activity during rewards for self, which is described in detail in the supplementary information. To unpack the win > lose contrasts for friends and self, we also examined the Friend win > Self win and Friend lose > Self lose contrasts."

Results section p. 15: "**Neural responses in NAcc when winning for friend > winning for self.** We also built up the model for winning for friend versus winning for self. This model building procedure showed no significant effects of age, sex, and friendship stability (see supplementary information).

Neural responses in NAcc when losing for friend > losing for self. The model building procedure with NAcc activity related to losing for friend versus losing for self, shows a negative effect of linear age on NAcc activity (left NAcc: random effects: SDintercept = .00, SDresidual = 2.17; fixed effects: [Intercept] b = .61, SE = .12, p < .001; [linear age] b = -.09, SE = .03 p < .01; right NAcc: random effects: SDintercept = .23, SDresidual = 2.45; fixed effects: [Intercept] b = 0.56, SE = .13, p < .001; [linear age] b = -.08, SE = .04 p = .03) (Figure S3). There were no effects of sex and friendship stability (see supplementary information for details)."

Supplementary information p. S3-S4: "**Neural responses in NAcc when winning for friend > winning for self.** We used the same model building procedure for left and right NAcc activity for the Friend win > Self win contrast. For left NAcc activity, sex did not explain additional variance to the model including linear and quadratic age terms (ps > .4), and neither did

friendship stability ($ps > .72$). The linear and quadratic age terms were also not significant ($ps > .28$). For right NAcc activity, there were also no effects of sex and friendship stability ($ps > .06$, and $> .78$, respectively), nor of the linear and quadratic age terms ($ps > .22$).

Neural responses in NAcc when losing for friend > losing for self. We first extended the null model with a linear age term, and second with a quadratic age term. These steps showed the best fit for models including a linear age term (left NAcc: $p = .0074$; right NAcc: $p = .03$, although for right NAcc only the AIC suggests a better fit above and beyond the null model). Above and beyond the linear and quadratic age terms, a main effect of sex or sex x age interaction effect did not improve the model fit ($ps > .86$ and $> .91$ for left and right NAcc respectively). Sex was therefore removed from the models. Extending the models with a main effect of friendship stability did also not improve the model fit ($p = .95$ and $p = .70$ for left and right NAcc, respectively). Although extending the models including a linear and quadratic age term and a main effect of friendship stability with age x friendship stability interaction terms significantly improved the model fit for left NAcc ($p = .03$), the AIC and BIC fit indices show that the model fit did not improve above and beyond a model including only a linear age term (see Tabel S2). For right NAcc, extending the model with age x friendship stability interaction terms did not improve the model fit ($p = .09$). Therefore, for both left and right NAcc a model including a linear age term was selected as the best fitting model, (left NAcc: random effects: $SD_{intercept} = .00$, $SD_{residual} = 2.17$; fixed effects: [Intercept] $b = .61$, $SE = .12$, $p < .001$; [linear age] $b = -.09$, $SE = .03$, $p < .01$; right NAcc: random effects: $SD_{intercept} = .23$, $SD_{residual} = 2.45$; fixed effects: [Intercept] $b = 0.56$, $SE = .13$, $p < .001$; [linear age] $b = -.08$, $SE = .04$, $p = .03$). These results show that across adolescence, NAcc activity related to losing for friends as compared to losing for self decreases with age (Figure S3 [A] for the fitted model, and [B] for the raw data)."

Discussion p. 19-20: "In this study, we tested whether having stable and unstable best friendships across a trajectory of four years in adolescence was associated with differential developmental trajectories of vicarious reward-related ventral striatum activity. When rewards were gained (vs. lost) for best friends, adolescents with stable best friendships showed a quadratic trajectory of change in ventral striatum activity, whereas adolescents with unstable best friendships showed no age-related changes in their ventral striatum responses to winning for their best friend. Consistent with prior research (Schreuders et al., 2018), winning for self resulted in a quadratic age pattern that was not different for the friendship groups. These effects were not found when directly contrasting winning for friends versus self, or for losing for friends versus self, suggesting that the effects are driven specifically by differential responses for winning versus losing. [...]"

When comparing winning with losing, our study showed that striatum activity in response to rewards for stable best friends (but not for unstable best friends) followed a peaking quadratic developmental trajectory. Additional analyses showed that participants with stable and unstable best friends showed similar age-related trajectories of ventral striatum activity when rewards were directed to the self (winning vs. losing for self) and when winning/losing for friends and winning/losing for self were compared. Interestingly, the effect of friendship stability on the vicarious reward-related contrast (i.e., winning vs. losing for friends) was not driven by

wins or losses, but mainly by the difference in reactivity to wins versus losses for best friends. These differential responses should be unpacked further in future studies by including several baseline conditions to examine within-person differences in responses to different feedback schemes further. Together, our results confirm our hypothesis that friendship stability is differentially associated with vicarious reward-related activity in the ventral striatum across development.”

Reviewer #3 (Remarks to the Author):

The authors have done a commendable job addressing previous concerns. I remain concerned about the rather ambiguous definitions of friendship stability, closeness and quality but I defer to the other reviewers and editor to make final judgment about whether that ambiguity limits interpretations.

We appreciate the reviewer’s in-depth comments and suggestions.

Reviewers' comments:

Reviewer #1 (Remarks to the Author):

I have no further comments.

Reviewer #2 (Remarks to the Author):

I thank the authors for adding the contrast of win for friend vs self. Unfortunately, the findings significantly dampen enthusiasm. Because there is no association between friendship stability and NAcc activation to friend vs self for winning or losing, results suggest that the NAcc to friends from the primary analysis is not specific to vicarious rewards. Given the manuscript (and title) focus on vicarious rewards, this is highly problematic.

Reviewer #2 (Remarks to the Author):

I thank the authors for adding the contrast of win for friend vs self. Unfortunately, the findings significantly dampen enthusiasm. Because there is no association between friendship stability and NAcc activation to friend vs self for winning or losing, results suggest that the NAcc to friends from the primary analysis is not specific to vicarious rewards. Given the manuscript (and title) focus on vicarious rewards, this is highly problematic.

We regret to see that the reviewer has interpreted the lack of effect of friendship stability on ventral striatum activity in *winning for friend vs. winning for self* as a lack of support for our conclusions. We firmly believe that our statistical approach confirms a relation between friendship stability and vicarious ventral striatum activity. Below, we outline three crucial points that support our conclusion. In short, we first show that our neuroimaging findings are robustly found in both the left and right ventral striatum. Second, we show that friendship stability relates to vicarious reward-related ventral striatum activity, but not to self-directed reward-related ventral striatum activity. Thirdly, we show that we followed a commonly-used statistical approach to examine reward-related activity. In addition, we ran post-hoc analyses with behavioral results that might help to understand the neuroimaging results.

Point 1: The goal of the current manuscript was to test whether developmental trajectories for neural responses to vicarious rewards are related to individual differences in friendship stability. To this end, we followed best friendship stability of a large sample of participants ($N=205$) across a period of five years and distinguished between participants with stable vs. unstable friendships ($n=123$), which resulted in a total of 346 scan sessions that were included in the analyses. This is a unique strength of the study. Our results showed that the developmental trajectory of ventral striatum activity for *winning vs. losing for best friends* across adolescence differed for participants with stable and unstable best friendships: ventral striatum activity followed a negative quadratic trajectory (i.e., with a peak) with age for participants with stable friendships, whereas there were no age-related changes for participants with unstable friendships. The effect of friendship stability on reward activity as function of age was found for both the left and right ventral striatum, supporting the robustness of this finding.

Point 2: The reviewer expressed the concern that the effect of friendship stability on changes in ventral striatum activity across adolescence may not be specific to vicarious rewards. To address this concern, we conducted a separate but similar set of analyses for *winning vs. losing for self*. Here, the results showed that friendship stability did not relate to the developmental trajectory of *winning vs. losing for self*. It is important to note that the latter finding is in line with previous studies showing a quadratic trajectory in ventral striatum responses to self-directed rewards across adolescence and adulthood. Together, the results showed that friendship stability related to *winning vs. losing for friends*, but not to *winning vs. losing for self*. We would like to thank the reviewer for pointing us in this direction because adding the *winning vs. losing for self* contrast has been a valuable contribution to the manuscript as it allows for using a broader perspective to interpret the relation between friendship stability and reward sensitivity.

Point 3: Furthermore, we would like to emphasize why we chose to examine the *winning vs. losing* contrasts (and not the *winning for friend vs winning for self* contrast as suggested by the reviewer). First, we followed the statistical approach of previous publications where reward sensitivity was defined as *winning vs. losing* (for self; e.g., Braams, van Duijvenvoorde, Peper, & Crone, 2015; Burani et al., 2019; Insel & Somerville, 2018; Schreuders et al., 2018). Hence, the *winning vs. losing* contrast extends previous work on neural reward sensitivity, where losing is often used a baseline. In fact, *neutral* is often experienced as losing and therefore it is the difference score that is most meaningful. Second, in our view *winning for friend vs. winning for self* (which could be rewritten as *winning for friend vs. self*) does not capture vicarious reward sensitivity, because the emphasis within this contrast is on the social target (friend versus the self) and does not capture the reactivity to the rewarding outcome (i.e., winning). In our opinion, these contrasts answer different research questions. The current study was aimed at examining reactivity to *winning* within the *friend* condition. Hence, we believe that reporting the *winning vs. losing for friend* and *winning vs. losing for self* is the best approach to achieve this goal. However, along with the reviewer's suggestion, we included the *winning for friend vs. winning for self* contrast in the final version of our manuscript. As we believe this contrast does not capture reward sensitivity when winning for friends well, we regret to see that the reviewer has interpreted the lack of effect of friendship stability on ventral striatum activity in *winning for friend vs. winning for self* as a lack of support for our conclusions.

Additional post-hoc analysis: We acknowledge the shortcomings of using “losing” as a baseline and elaborated on this point in our discussion. As we do not have a more neutral baseline condition to include in the brain contrasts, we propose to provide a more comprehensive overview of self-reported pleasure from winning and losing in the manuscript. Currently, we examine self-reported ratings of *pleasure from winning vs. losing* in the manuscript. We chose to use this difference score of *winning* and *losing* to match the neural contrast of *winning vs. losing* (also in line with Schreuders et al., 2018). We now also ran post-hoc analyses to explore the effect of friendship stability on the developmental trajectory of pleasure from winning and pleasure from losing separately (i.e., *pleasure from winning for friend*, and *pleasure from losing for friend*). In short, these analyses show no effects of friendship stability on age-related changes of pleasure from winning and losing for friend. Together with the lack of significant effects of friendship stability on ventral striatum activity for *winning for friend > winning for self* and *losing for friend > losing for self*, one might argue that the relative difference of winning and losing plays an important role in our findings. (However, it should be noted that pleasure from winning vs. losing did not yield significant relations with friendship stability). We now report these additional post-hoc analyses and results in the supplementary materials and discuss them in the manuscript (changes are highlighted in blue).

For example, see the Discussion:

“When comparing winning with losing, our study showed that striatum activity in response to rewards for stable best friends (but not for unstable best friends) followed a peaking quadratic developmental trajectory. Additional analyses showed that participants with stable and unstable best friends showed similar age-related trajectories of ventral striatum activity when rewards were directed to the self (*winning vs. losing for self*) and when *winning/losing for friends* and *winning/losing for self* were compared. Interestingly, the effect of friendship stability on the vicarious reward-related contrast (i.e., *winning vs. losing*

for friends) was not driven by wins *or* losses (as examined with neural contrasts winning for friend > winning for self, and losing for friend > losing for self), but mainly by the difference in reactivity to wins versus losses for best friends. These differential responses should be unpacked further in future studies by including several baseline conditions to examine within-person differences in responses to different feedback schemes further. Together, our results confirm our hypothesis that friendship stability is differentially associated with vicarious reward-related activity in the ventral striatum across development.” (P. 20-21)

[...]

“Some limitations should be acknowledged. We acknowledge the shortcomings of using “losing” as a baseline condition when examining vicarious reward responses of the ventral striatum, although *winning vs. losing* is a commonly used contrast to examine reward sensitivity (for self; e.g., Braams, van Duijvenvoorde, Peper, & Crone, 2015; Burani et al., 2019; Insel & Somerville, 2018; Schreuders et al., 2018). In an attempt to examine reward sensitivity (i.e., when winning) and “punishment” sensitivity (i.e., when losing) in isolation, we used the self-reported pleasure ratings to conduct post-hoc analyses. We explored the effect of friendship stability on the developmental trajectory of pleasure from winning and losing for friends and self separately. Our findings show no effects of friendship stability on pleasure from winning or losing for friends. Tentatively, these null findings may also highlight the important role of the relative difference of reactivity to winning and losing.” (P. 23)

For example, see the methods and results section:

“Post-hoc, we also examined the pleasure ratings for pleasure from winning for friend (ICC = .62, 95%-CI = [0.52, 0.70]) and pleasure from losing for friend separately (ICC = .23, 95%-CI = [0.03, 0.39]). The results are reported in the supplementary information and briefly discussed in the manuscript.” (P. 10)

“Post-hoc, we also explored the effect of pleasure from winning and losing for friends separately (i.e., pleasure from winning for friend, and pleasure from losing for friend). These results are described in the supplementary information. In short, these analyses show no effects of friendship stability on age-related changes of pleasure from winning and losing for friend.” (P. 16)

In addition, we more extensively discuss the behavioral results pertaining the friendship characteristics:

“We also examined the role of friendship stability on developmental trajectories of friendship closeness, friendship quality, and pleasure from vicarious winning vs. losing. Those participants who reported to have unstable best friendships showed an age-related decrease in closeness with the current best friend extending into adulthood. With a longitudinal, behavioral study on friendship stability, Bowker, Rubin, and Burgess (2006) show that young adolescents (here: around the age of 10 years) with stable best friendships were socially equally well-adjusted as adolescents with unstable best friendships. This highlights the importance of having any best friend in early adolescence. The current study extends these findings by showing that friendship closeness become more significant across adolescence, and emphasize that stable friendships may have long-term positive effects on

developing close relationships. Furthermore, we showed, for males only, that best friendship quality related to friendship stability such that positive friendship quality was higher for adolescents with stable best friendships than for adolescents with unstable best friendships. There were no differences in pleasure from winning (vs. losing) for the best friend for participants with stable and unstable best friendships. Our findings support previous notions that adolescents and young adults with stable and unstable best friendships differ in their friendship characteristics (Poulin & Chan, 2010).” (P. 21-22)

REVIEWERS' COMMENTS:

Reviewer #1 (Remarks to the Author):

The authors have done a good job with the revision. Adding the new post-hoc analyses and inclusions in the discussion about using the loose trials as a baseline, along with including all the analyses for transparency, strengthens the paper. Together with the the timeliness of the topic and ambitious longitudinal design will be well-received by readers of Nature Communication.

Reviewer #2 (Remarks to the Author):

the authors have addressed my concerns